

# Stable isotopes reveal evaporation dynamics at the soil-plant-atmosphere interface of the critical zone

Matthias Sprenger, Doerthe Tetzlaff, Chris Soulsby

Northern Rivers Institute, School of Geosciences, University of Aberdeen, Aberdeen, AB24 3UF, UK

*Correspondence to*: Matthias Sprenger (Matthias.sprenger@abdn.ac.uk)

## Abstract

Understanding the influence of vegetation on water storage and flux in the upper soil is crucial in assessing the consequences of climate and land use change. We sampled the upper 20 cm of podzolic soils at 5 cm intervals in four sites differing in their vegetation (Scots Pine (*Pinus sylvestris*) and heather (*Calluna sp. and Erica Sp*)) and aspect.

The sites were located within the Bruntland Burn long-term experimental catchment in the Scottish Highlands; a low energy, wet environment. Sampling took place on 11 occasions between September 2015 and September 2016 to capture seasonal variability in isotope dynamics. The pore waters of soil samples were analysed for their isotopic composition ($\delta^2H$ and $\delta^{18}H$) with the direct equilibration method. Our results show that the soil waters in the top soil are, despite the low potential evaporation rates in such northern latitudes, kinetically fractionated compared to the

precipitation input throughout the year. This fractionation signal decreases within the upper 15 cm resulting in the top 5 cm being isotopically differentiated to the soil at 15 – 20 cm soil depth. There are significant differences in the fractionation signal between soils beneath heather and soils beneath Scots pine, with the latter being more pronounced. But again, this difference diminishes within the upper 15 cm of soil. The enrichment in heavy isotopes in the topsoil follows a seasonal hysteresis pattern, indicating a lag time between the fractionation signal in the soil and the

increase/decrease of soil evaporation in spring/autumn. Based on the kinetic enrichment of the soil water isotopes, we estimated the soil evaporation losses to be about 5 and 10 % of the infiltrating water for soils beneath heather and Scots pine, respectively. The high sampling frequency in time (monthly) and depth (5 cm intervals) revealed high temporal and spatial variability of the isotopic composition of soil waters, which can be critical, when using stable isotopes as tracers to assess plant water uptake patterns within the critical zone or applying them to calibrate tracer-

aided hydrological models either at the plot to the catchment scale.

## 1. Introduction

Processes in the soil-plant-atmosphere continuum exert a major influence on water partitioning into evaporation, transpiration and recharge fluxes. Therefore, the outer part of the Earth's terrestrial surface, where the subsurface is closely coupled with the atmosphere and vegetation, is often referred to as the critical zone (Brooks et al., 2015).

However, the dynamics of feedbacks between soils and vegetation remain insufficiently well understood (Werner and Dubbert, 2016). Consequently, there is an increased interest in improving the conceptualization of the upper boundary of soils, as the important interface between soils-plant-atmosphere. For example, it has been shown that evapotranspiration dynamics can affect travel times of percolating water in the unsaturated zone (Sprenger et al.,





2016c; Heße et al., 2017) and catchment outflows (van der Velde et al., 2015; Rinaldo et al., 2011). Additionally, understanding the age distributions of evapotranspiration fluxes themselves has recently gained interest in the literature (Harman, 2015; Soulsby et al., 2016a; van Huijgevoort et al., 2016; Queloz et al., 2015). However, estimates of these evapotranspiration ages and catchment travel times require a sound understanding of the storage and mixing

dynamics of the subsurface and surface water pools which form the sources of evapotranspiration within catchments. Disentangling these atmospheric losses into evaporation and transpiration is particularly challenging.

Stable isotopes of water ($^2$H and $^{18}$O) are a powerful tool for the analysis of the partitioning of water (see review by Kool et al., 2014). They are often seen as ideal tracers, since they are part of the water molecule. During root water uptake, the isotopic composition of the remaining soil water is usually not altered (Wershaw et al., 1966; Dawson and

Ehleringer, 1991). However, evaporation is a fractionating process, where the remaining water is generally enriched in heavy isotopes (equilibrium fractionation). Additionally, in natural open systems with a humidity of < 100 %, $\delta^2$H is more likely to be evaporated than $\delta^{18}$O, because of their different atomic weights, leading to kinetic non-equilibrium fractionation (Craig et al., 1963). Therefore, evaporation losses result in an isotopic signal in the residual water that is distinct from the original isotopic composition of the precipitation waters that were formed in isotopic equilibrium

(Dansgaard, 1964).

Such enrichment from kinetic fractionation was found in soil water isotopes across various climatic regions; with more pronounced evaporative signals reaching deeper into the soils in arid and Mediterranean environments than in temperate regions (Sprenger et al., 2016b). So far, soil water isotopes have been studied much less extensively in the colder regions of the northern latitudes (but see Tetzlaff et al., 2014; Geris et al., 2015b; Geris et al., 2015a). However,

the knowledge of soil water isotopic composition is of fundamental importance when studying the root water uptake pattern of plants. While comparisons of the isotopic signal in soil waters with waters of plant tissues have been reported for decades (see review by Ehleringer and Dawson, 1992), recent technical developments have enabled easier analysis of stable isotopes in soil water (see review by Sprenger et al., 2015a). Recently posed research questions on which water of the soils pore system is used by plants (Brooks et al., 2010) are enhancing ecohydrological studies on soil –

plant interactions in the critical zone (Evaristo et al., 2016; Volkmann et al., 2016; McCutcheon et al., 2016; Hervé-Fernández et al., 2016; Oerter and Bowen, 2017).

A comparative study by Evaristo et al. (2015) showed that the isotopic composition of plant waters - just like soil waters in the upper horizons – are usually kinetically fractionated. However, Evaristo et al. (2015) did not cover the northern latitudes in their review, since there has been only one preliminary study, looking into the root water uptake

of the vegetation in this low energy region (Geris et al., 2015a). Geris et al. (2015a) found relatively little fractionation in the soil waters at -10 cm soil depth and xylem waters in the Bruntland Burn catchment in Northern Scotland. However, recent findings about kinetic fractionation in the water pools and tracks of an extended drainage network in a raised bog within the Bruntland Burn showed that despite the low energy environment, evaporation can have a fractionating effect on the stable isotopes of peatland waters (Sprenger et al., 2017b). While such isotopic fractionation

of open waters in peatlands of the northern latitude were found by others (Carrer et al., 2016; Gibson et al., 2000), the potential fractionation of the water in the upper soil layer in these cold regions have not previously been studied.





Thus there is a particular need to better understand such ecohydrological processes in the higher latitudes (Tetzlaff et al., 2013).These environments are known to be especially sensitive to future climate change projections, since relatively little warming could cause intense changes in the water balance when snowfall and snow melt dynamics (Mioduszewski et al., 2014) and vegetation phenology shift (Shen et al., 2014). The upper soil layers (top 30 cm), where about 90 % of the root mass is usually present in the Northern temperate and boreal biomes (Jackson et al., 1996), are of special interest to better understand how vegetation influences the partitioning of soil water into evaporation, transpiration and recharge. The marked seasonality in the northern environments and its impact on the evaporation signal in soil water stable isotopes are fertile areas for investigation.

Here, we address the following research questions in order to improve understanding of the evaporation dynamics at the soil-plant-atmosphere interface and their influences on the water storage and mixing in the critical zone:

How do precipitation input and the critical zone water storage mix and affect the soil water isotope dynamics over time?

How can one derive soil evaporation dynamics from soil water isotopic fractionation?

How does the soil-plant-atmosphere feedbacks vary in space, depending on the soil, vegetation, and aspect?

## 2. Methods and study sites

### 2.2 Environmental conditions

Our study was conducted in the Bruntland Burn (BB) experimental catchment (3.2 km$^2$) in the Scottish Highlands; a sub-catchment of the Girnock burn, a long-term ecohydrological research site. A detailed description of the soil and vegetation characteristics follows in section 2.3. The underlying geology of the BB is characterized by granitic and metamorphic rocks (Soulsby et al., 2007). About 60 % of the catchment are covered by up to 40 m of glacial drift deposits which maintain a high groundwater storage (Soulsby et al., 2016b). The climate is temperate/boreal oceanic with mean daily air temperatures ranging between 2°C in January and 13°C in July. Annual precipitation is about 1000 mm yr$^{-1}$, which is fairly evenly distributed and occurs mainly as rainfall (usually < 5 % as snow) of low intensities (50 % of rainfall at intensities of < 10 mm d$^{-1}$) (Soulsby et al., 2015). The annual potential evaporation (PET) is about 400 mm yr$^{-1}$ and the annual runoff is around 700 mm year$^{-1}$. While the runoff of the BB shows limited seasonality, though lower flows tend to be in summer, the PET estimated with the Penman-Monteith approach follows a strong seasonal dynamic with average PET rates of 0.3 to 0.7 mm d$^{-1}$ from November to February and 2.3 to 2.7 mm d$^{-1}$ from May to August (Sprenger et al., 2017b).

The seasonality of the climate is also reflected in the variability of the isotopic signal of the precipitation with depleted values being more common during winter (dropping frequently below − 80 ‰ $\delta^2$H from November to March) and more enriched values dominant in summer. The weighted averages for the precipitation isotope signal over 5 years (2011-2016) were $P_{avg}$ $\delta^{18}$O = -8.5 ‰ and $P_{avg}$ $\delta^2$H = -61 ‰. The regression (calculated via scipy.stats.linregress in



Python) between $\delta^{18}O$ and $\delta^2H$ values of daily precipitation data sampled between June 2011 and September 2016 describes the local meteoric water line (LMWL):

$$\delta\ ^2H = 7.6 \times \delta^{18}O + 4.7 \tag{1}$$

**2.3  Study sites**

Soil sampling focused on four different sites within the BB, where the soils are characterized as freely draining podzols (Figure 1a). The study sites differed with regard to their vegetation cover and their aspect: At two sites, Scots Pine (Pinus sylvestris) forest is the dominant vegetation and at the two other sites, heather (*Calluna sp.* and *Erica sp.*) shrubland is dominating (Figure 1b). Each of the two soil-vegetation landscape units were studied at a north facing
and a south facing slope (sites had gentle slopes), leading to the following four different sites: North facing heather (NH), north facing forest (NF), south facing heather (SH), and south facing forest (SF).

The podzols are shallow soils and frequent large clasts within the glacial drift deposit usually inhibit soil sampling below 20 to 30 cm. The soils at the four study sites were relatively similar with regard to their colour and texture (Figure 1c). The texture of the upper 20 cm was determined for each site in 5 cm depth increments. After ignition of
the soil samples to free the soil of organic content, the coarse and medium sand fractions were determined by dry sieving. The fine sand, silt, and clay fractions were estimated with the hydrometer method (Gee and J. W. Bauder, 1986). The upper 20 cm consist mainly of loamy sand and only for SH, the top 10 cm indicated a higher contribution of the silt fraction leading to sandy loam (Table 1). For NH and SH, the gravel content as well as the sand content generally increases with depth. The texture analysis proved infeasible for the upper 5 cm at SH, NF, and SF, since
there was not enough soil material left after ignition of the organic material. Coarse gravel (> 20 mm diameter) were only present in the soil below 5 cm at SF, where also the highest fine gravel content was also present. The bulk density of the podzols in the BB is about 0.74 g cm$^{-3}$ for the top 20 cm (Geris et al., 2015b).

The organic content in the upper 20 cm of the soils was determined for each study site in replicated 5 cm depth increments (n = 5 per site and depth) by loss on ignition (LOI) of about 10 g soil material at 550°C in a furnace over
2h according to Ball (1964). The LOI decreased linearly with soil depth at all study sites. Pearson correlations between LOI and soil depth were strong and significant at the 99 % confidence with NH, SH, and SF and weaker for NF (Table 1). The LOI showed generally a strong relationship with the gravimetric water content (generally r ≥ 0.7, p < 0.01) at all four study sites (Table 1). We determined the gravimetric water content (GWC) of all soil samples by relating the weight loss after oven drying at 105 °C over night to the dry soil mass.

Hemispheric photos taken during the vegetation period revealed that the median of the canopy coverage (CC) of the heather was slightly lower (NH: CC = 65 %, n = 9; SH: CC = 62 %, n = 9) than for the forested sites (NF: CC = 67 %, n = 36; SF: CC = 69 %, n = 46) (Braun, 2015). The forested sites were both plantations, but with larger trees (mean diameter at breast height (DBH) = 21.8 cm) and lower tree density at SF compared to NF (mean DBH = 13.8 cm),





where the tree ages were more variable (Braun, 2015). The height of the heather vegetation was limited to about 0.5 m and tree height was 12 – 15 m in the plantations.

Fine root (defined as 0.5 – 2 mm according to Zobel and Waisel (2010)) density was determined by wet sieving of the fine roots from soil cores (100 cm³) taken from the upper 20 cm in 5 cm increments. Afterwards, the roots were dried

at 70 °C overnight. The root density was then derived by relating the dried root mass at each depth to the total root mass in the profile. This analysis was limited to the heather sites, since the roots and boulders at the forested sites inhibited sampling of undisturbed soil cores. The roots of heather were limited to the upper 15 cm at the heather sites with almost exponential decrease at NH and a more linear decrease at SH (Table 1).

### 2.4  Sampling design and analysis

Soil sampling at each site was conducted at monthly intervals between September 2015 and September 2016 (except for Dec., Feb., March; n = 11). For each sampling campaign, soils at all four sites were sampled with a spade across five profiles in 5 cm increments down to 20 cm soil depth. For each sampling depth, five replicate samples were taken to account for the high subsurface heterogeneity (Figure S 1). Each soil sample contained of about 80 to 250 g of soil and was stored in air tight bags (Weber Packaging, Güglingen, Germany), ensuring - by manually furling the bags -

that as little air as possible was inside them. For the sampling campaign in May, five additional sites with heather vegetation were sampled on the north (n=2) and south facing (n=3) slopes to increase the sample size for comparisons between the two slopes and get an idea of the general variability in space. In addition to the monthly sampling, two extra sampling campaigns for the two heather sites were conducted in August (4th and 9th August 2016) to investigate short time changes on both slopes and to further increase the sample number for comparison between the two slopes.

The soil samples were analysed for their stable isotopic composition ($\delta^2H$ and $\delta^{18}O$) according to the direct equilibration method suggested by Wassenaar et al. (2008). For a detailed description of the soil water isotope analyses in the lab of the Northern Rivers Institute at the University of Aberdeen with the direct equilibration method using off-axis Integrated Cavity Output Spectroscopy (OA-ICOS) (triple water-vapour isotope analyser TWIA-45-EP, Model#: 912-0032-0000, Serial#: 14-0038, Manufactured: 03/2014, Los Gatos Research, Inc., San Jose, CA, USA),

we refer to Sprenger et al. (2017a). To assess the precision of the analysis, we derived the standard deviation of in total 81 measurements of a standard water (Aberdeen tap water) sampled along with the soil samples at the beginning, the middle and the end of each of the 27 days of laboratory analyses over one year. The standard deviation of the standard water analysis was 0.31 ‰ for $\delta^{18}O$ values and 1.13 ‰ for $\delta^2H$ values. Recently reported potential effects of $CO_2$ on the isotope analysis of vapour with wavelength-scanned cavity ring-down spectroscopy (Gralher et al., 2016)

have been shown to not apply to the OA-ICSO that we used, as shown by Sprenger et al. (2017a).

In addition to the soil water analysis, precipitation at the field site was sampled with an auto sampler on daily basis at the catchment outlet (location shown in Figure 1a). The auto sampler was emptied at least every two weeks and evaporation from the sampling bottles was prevented by adding paraffin. The precipitation isotopic composition ($\delta^2H$ and $\delta^{18}O$) was determined with the above mentioned OA-ICOS running in liquid mode with a precision of 0.4 ‰ for

$\delta^2H$ values and 0.1 ‰ for $\delta^{18}O$ values, as given by the manufacturer.





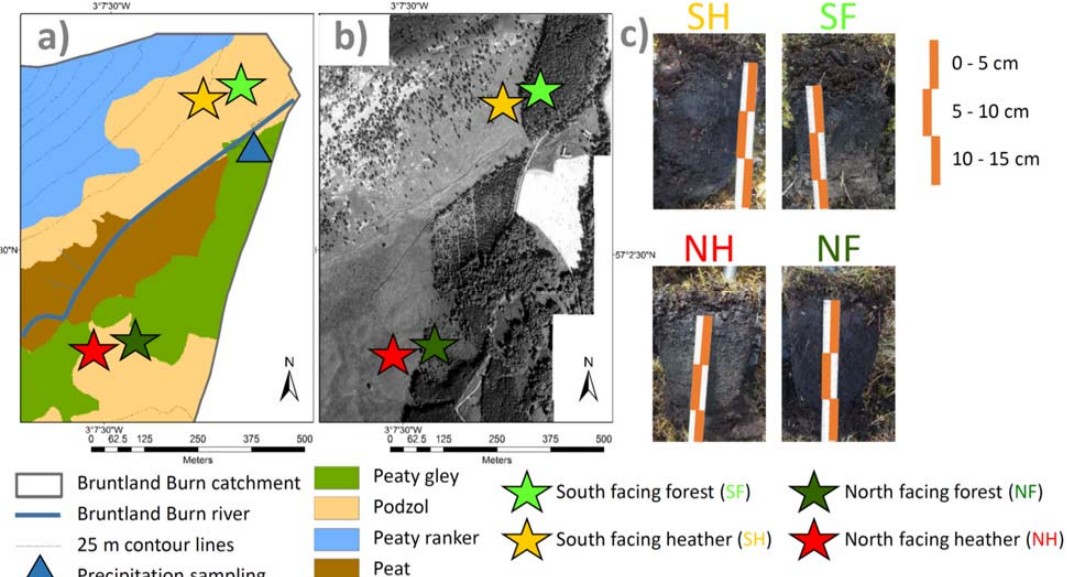

**Figure 1 Location of the four sampling sites within the Bruntland Burn catchment on (a) a soil map and (b) an aerial photo. The precipitation sampling location is indicated by a blue triangle in (a). (c) The four photos on the right show exemplary soil profiles for the four study sites.**

### 2.5 Data analysis

We calculated the evaporation line (EL) as a regression line through the soil water isotope data of each sampling date in the dual isotope space (scipy.stats.linregress in python). The EL is characterized by its slope and intercept with the $\delta^2$H axis. All regressions for EL presented here were significant at the 95 % confidence interval. For each soil water and precipitation sample, we further calculated the line conditioned excess (lc-excess) as a function of the slope ($a = 7.6$) and the intercept ($b = +4.7$ ‰) of the LMWL (Equation 1) as suggested by Landwehr and Coplen (2006):

$$lc - excess = \delta\,^2H - a \times \delta^{18}O - b \qquad (2)$$

The lc-excess describes the deviation of the sample's $\delta^2$H value the LMWL in the dual isotope space (Landwehr et al., 2014), which indicates non-equilibrium kinetic fractionation processes due to evaporation after precipitation. Therefore, the lc-excess is similar to the well-established deuterium-excess (Dansgaard, 1964) that relates the deuterium composition to the global meteoric water line (GMWL). However, we found that lc-excess was advantageous over the deuterium-excess (or single isotope approaches with $\delta^2$H or $\delta^{18}$O) for inferring evaporation fractionation, because the lc-excess of the precipitation input is about 0 ‰ and with relatively little seasonal dynamics, while $\delta^2$H, $\delta^{18}$O, and d-excess can have an intense seasonal variability (Sprenger et al., 2017b).

To infer dynamics of potential evaporation rates, we estimated potential evapotranspiration (PET) with the Penman-Monteith-Equation adjusted for the Scottish Highlands by Dunn and Mackay (1995). Note that we focus in our study on the PET dynamics and that the absolute values could vary depending on the aerodynamic and roughness parameter





of different vegetation covers. We further did not partition PET into evaporation and transpiration fluxes, since PET was primarily used as a proxy for potential soil evaporation rates, and evaporation and transpiration usually show a linear relationship in temperate regions (Renner et al., 2016; Schwärzel et al., 2009). To investigate the effect of antecedent conditions, average values of PET over 30 days prior to each sampling campaign were calculated ($PET_{30}$).

Additionally, the precipitation sums and the weighted isotopic signal of the precipitation over 7 days and 30 days prior to the sampling was calculated ($P_7$ and $P_{30}$, respectively).

We estimated the evaporative water losses $f$ and evaporation/input ratios ($E/I$) based on the Craig-Gordon model (Craig and Gordon, 1965) with the *Hydrocalculator* provided by Skrzypek et al. (2015). For a detailed description of the calculations, we refer to Skrzypek et al. (2015) and will only briefly introduce the main characteristics of the approach

and assumptions we used in our application. The estimate of the evaporative losses $f$ [%] is based on formulations introduced by Gonfiantini (1986) and adapted in the *Hydrocalculator* for isotope mass balance as follows:

$$f = 1 - \left[ \frac{(\delta_S - \delta^*)}{(\delta_P - \delta^*)} \right]^m \tag{3}$$

For our estimates of $f$, with defined $\delta_S$ as the isotopic signal of the soil water in the upper 10 cm. The upper 10 cm were chosen, because this was the depth with the highest evaporation signal in the soil water isotopes, as shown in the

results section. The sampled soil water isotope signal $\delta_{SW}$ was corrected for the seasonality of the input signal by the average isotopic signal of $P_{30}$ as $\delta_S = \delta_P + (\delta_{SW} - \delta_{P30})$, $\delta_P$ as the isotopic signal in the long term precipitation input (-61 ‰ for $\delta^2H$ and -8.5 ‰ for $\delta^{18}O$), $\delta^*$ as the limiting isotopic enrichment factor, and $m$ as the enrichment slope. $\delta^*$ is a function of the air humidity $h$, the isotopic composition of the ambient air $\delta_A$, and a total enrichment factor $\varepsilon$ (Gat and Levy, 1978). For humidity, we averaged over 30 days prior to each soil water sampling date the measured humidity

at a meteorological station less than 800 m away from the study sites $h_{30}$. $\delta_A$ was derived as function from the weighted average precipitation input of the 30 days prior to the soil sampling $\delta_{P30}$ and the equilibrium isotope fractionation factor $\varepsilon^+$, which depends on the temperature as given by Horita and Wesolowski (1994). We used air temperature data from the aforementioned meteorological station and computed values averaged over 30 days prior to soil water sampling $T_{30}$. $\varepsilon$ is the sum of the equilibrium isotope fractionation factor $\varepsilon^+$ and the kinetic isotope fractionation factor

$\varepsilon_k$, which is a function of $h_{30}$ and kinetic fractionation constant (Gonfiantini, 1986). The enrichment slope $m$ was calculated as a function from $h_{30}$, $\varepsilon$, and $\varepsilon_k$ in accordance to Welhan and Fritz (1977).

The ratio between evaporated water $E$ [%] to the net precipitation input $I$ can be calculated with the *Hydrocalculator* in accordance to Allison and Leaney (1982) as:

$$E/I = \left[ \frac{(\delta_S - \delta_P)}{(\delta^* - \delta_S) \times m} \right] \tag{4}$$

Both $E/I$ and $f$ can be derived either by $\delta^2H$ or $\delta^{18}O$, indicated as $E/I_{d2H}$, $f_{d2H}$, $E/I_{d18O}$, and $f_{d18O}$. Note that our approach is based on the assumption that 1.) the soil evaporation mainly happens from the soil water in the upper 10 cm (justified by the observations given in the results section), 2.) that isotopic enrichment of soil waters was defined as the enrichment compared to the long term isotopic signal of the precipitation input, and 3.) that the averages over 30 days





prior to the soil water sampling for humidity ($h_{30}$), air temperature ($T_{30}$), and precipitation isotopes ($\delta_{P30}$) represent the atmospheric drivers for the isotopic enrichment. Note that $\delta_{P30}$ represents in these calculations the net precipitation infiltrating the soil, which has the same isotopic composition as the measured rainfall isotope values, since no evidence for isotopic enrichment was found for the throughfall and stem flow in heather and Scots pine stands in the Bruntland

Burn catchment (Braun, 2015).

Statistical analyses for the soil water isotopes ($\delta^2H$, $\delta^{18}O$ and lc-excess) for individual sites, depths, and dates were done with non-parametric tests, since the null-hypothesis that the data was drawn from a normal distribution was rejected for several sampling campaigns using the Shapiro-Wilk test for normality (scipy.stats.shapiro.test in Python). We also tested whether there were significant differences between the sites (each site with n > 200) and sampling

dates (each date with n > 75) or at different depths (each depth with n > 200) with the Kruskal-Wallis test (kruskal.test in R). When significant differences were present at the 95 % confidence interval, a post-hoc Dunn test (posthoc.kruskal.dunn.test in R) with p-value adjustment "bonferroni" was applied to see which of the sampling dates or sampling depths were significantly different. For pairwise tests (e.g., aspect (north- and south-facing) or vegetation (soils beneath heather and soils beneath Scots pine; n > 35 for each vegetation type on each sample day and n > 100

for each vegetation type at each sample depth), the nonparametric Mann-Whitney-Wilcoxon test was used (pairwise.wilcox.test in R). We assessed if the soil water lc-excess values for different sites were significantly lower than 0, -1, -2, -3, -4, -5, -6, an -7 ‰ by using the Wilcoxon signed-rank test (scipy.stats.wilcoxon in Python).

Mean values for each of the 11 sampling dates of $\delta^2H$, $\delta^{18}O$, and lc-excess of soil water and $P_7$ or GWC, $PET_{30}$, $P_7$, and the lc-excess of $P_{30}$ were normally distributed according to the Shapiro-Wilk test. Therefore, we used the T-test

for the mean of one group of samples (scipy.stats.ttest_1samp in Python) to test if the average soil water lc-excess deviated significantly from zero for any sampling campaign. We further calculated the Parson correlation coefficients (r) to describe linear relationships between mean values (scipy.stats.pearsonr in Python). Since $\delta^2H$, $\delta^{18}O$ of $P_{30}$ were not normally distributed, the Spearman rank correlation ($\rho$) was applied to describe relationships with these two variables (scipy.stats.spearmanr in Python). For all statistical analyses, the 95 % confidence interval was defined as

significance level (p < 0.05). Visualizations with boxplots generally show the interquartile range (IQR) as boxes, the median as line within the box, the 1.5 IQR as whiskers and data points 1.5 IQR as points (matplotlib.boxplot in Python). We further make use of violin plots, where a kernel density estimation of the underlying distribution describing the data is visualized (seaborn.violinplot in Python). Statistical differences derived with the post-hoc Dunn test are visualized by either letters or coloured markers within the boxplots, where the same letter or marker color

indicate that the samples are not significantly different to each other. Statistical differences between two groups derived with the Mann-Whitney-Wilcoxon test are indicated by either an asterisk or "X".




## 3 Results

### 3.1 Temporal dynamics in soil water isotopes

The temporal dynamic of the seasonally variable precipitation input signal between September 2015 and September 2016 was generally imprinted on the soil water (SW) isotope data at all sites. $P_{30}$ $\delta^2H$ values usually were most similar
to the lowest (most depleted) SW $\delta^2H$ values (Figure 2b). An exception to this were the soil samples from January, because the sampling took place after a period of intense rainfall ($P_{30}$ = 434 mm) that occurred at the end of December and beginning of January (Figure 2a). The SW $\delta^2H$ (and $\delta^{18}O$) values averaged over all sites and depths for each sampling campaign correlated significantly with $P_{30}$ $\delta^2H$ ($\rho = 0.92$, $p < 0.01$) and $P_{30}$ $\delta^{18}O$ ($\rho = 0.97$, $p < 0.01$) and $P_7$ $\delta^2H$ ($r = 0.74$, $p = 0.01$) and $P_7$ $\delta^{18}O$ ($r = 0.83$, $p < 0.01$). However, the soil water averages were usually more enriched
than the precipitation input (except for the January sampling). There is relatively little variability in the P lc-excess, which had a weighted average of -0.58 ‰. The average SW lc-excess was usually more negative than the $P_{30}$ lc-excess, but the sampling campaigns on 13 January and 18 March in 2016 were an exception (Figure 2c, Table 2). In contrast to $\delta^2H$ and $\delta^{18}O$, there was neither a relationship between the SW lc-excess with $P_{30}$ lc-excess ($r = 0.33$, $p = 0.32$) nor with $P_7$ lc-excess ($r = 0.18$, $p = 0.59$). Thus, the dynamics of SW lc-excess cannot be explained by variation
of the lc-excess in the input.

The seasonally variable input of P $\delta^2H$ and $\delta^{18}O$ led to significantly more depleted SW isotope values during January and March compared to the other sampling days (indicated by the letter "a" at the boxplots in Figure 3). The sampling in November and April represented a transition period, where the SW isotopic composition was significantly different to the Winter and Summer samples. The SW $\delta^2H$ and $\delta^{18}O$ values between May and August did not differ significantly.
The soil water samples from January were the only ones that plotted along the LMWL with a slope of the EL of 8.2 (cyan dots in Figure 3). The other soil water samples followed ELs of slopes between 3.7 and 5.4 with the lowest slopes at end of summer and beginning of autumn (Table 2). The variability of $\delta^2H$ and $\delta^{18}O$ values was highest for the sampling during winter and generally higher for $\delta^{18}O$ compared to $\delta^2H$ (note that the axes are scaled according to the GMWL in Figure 3).

The SW isotopes were usually more enriched towards the soil surface. Only the $\delta^2H$ depth profiles for the November and January sampling showed more depleted values in the topsoil (Figure 4a). The lc-excess depth profiles revealed that the deviation from the LMWL decreased with soil depth for all sampling campaigns (Figure 4b). While the values varied over the year, with more positive lc-excess values in winter, the shape of the lc-excess profiles remained relatively persistent throughout the year. In addition, the GWC across the upper 20 cm of soil had a fairly persistent
pattern over the year with generally highest values in the top 5 cm and decreasing values with soil depth (Figure 4c).

The enrichment in SW $\delta^2H$ in the upper 20 cm of the soil during spring (green dots in Figure 5a) followed an increase in $PET_{30}$. Highest $PET_{30}$ in summer correspond with enriched $\delta^2H$ values. While $PET_{30}$ decreased at the end of summer, the SW stayed enriched in $^2H$ until late autumn (orange dots in Figure 5a). Thus, the SW $\delta^2H$ (and also $\delta^{18}O$, not shown) to $PET_{30}$ relationship can be described by a linear correlation ($r = 0.65$, $p = 0.03$ for $\delta^2H$ and $r = 0.64$, $p =$





0.03 for $\delta^{18}O$). However, Figure 5a shows that the delayed response in the SW isotopic signal to changes in $PET_{30}$ resulted in a hysteresis pattern. The error bars in Figure 5a, representing standard deviations of all the samples for each sampling campaign, show again that the SW $\delta^2H$ values were most variable for the sampling in January, becoming less variable during spring and lowest in autumn.

The hysteresis pattern was very pronounced for the relationship between SW lc-excess and $PET_{30}$, since the SW lc-excess only increased little with onset of $PET_{30}$ in spring time (green dots in Figure 5b). During summer, $PET_{30}$ remained high, and SW lc-excess values became lower and stayed low, even when $PET_{30}$ started to decrease (Figure 5b). However, the relationship between SW lc-excess and $PET_{30}$ could statistically be also described by a linear relationship ($r = -0.64$, $p = 0.04$). Note that since there was no relationship between SW lc-excess and $P_{30}$ lc-excess,

the SW lc-excess to $PET_{30}$ relationship also showed a hysteresis pattern when applying the correction for seasonality using the difference between SW lc-excess and $P_{30}$ lc-excess as described in the methods. It is further worth noting that the average SW lc-excess in the upper 20 cm was for all sampling campaigns, apart from the January sampling, significantly < 0 ‰, (T-test, $p = 0.63$). When limiting the analysis to samples from the upper 5 cm, there was also a hysteresis pattern in the relationship between SW lc-excess and $PET_{30}$.

The average GWC for each sampling campaign correlated significantly with the average SW lc-excess for that sampling day, with lower SW lc-excess when soils were drier ($r = 0.67$, $p = 0.02$). The GWC also correlated with SW $\delta^{18}O$ ($r = -0.75$, $p = 0.01$) and $\delta^2H$ ($r = -0.74$, $p = 0.01$) with isotopically more enriched values when the soil was drier.

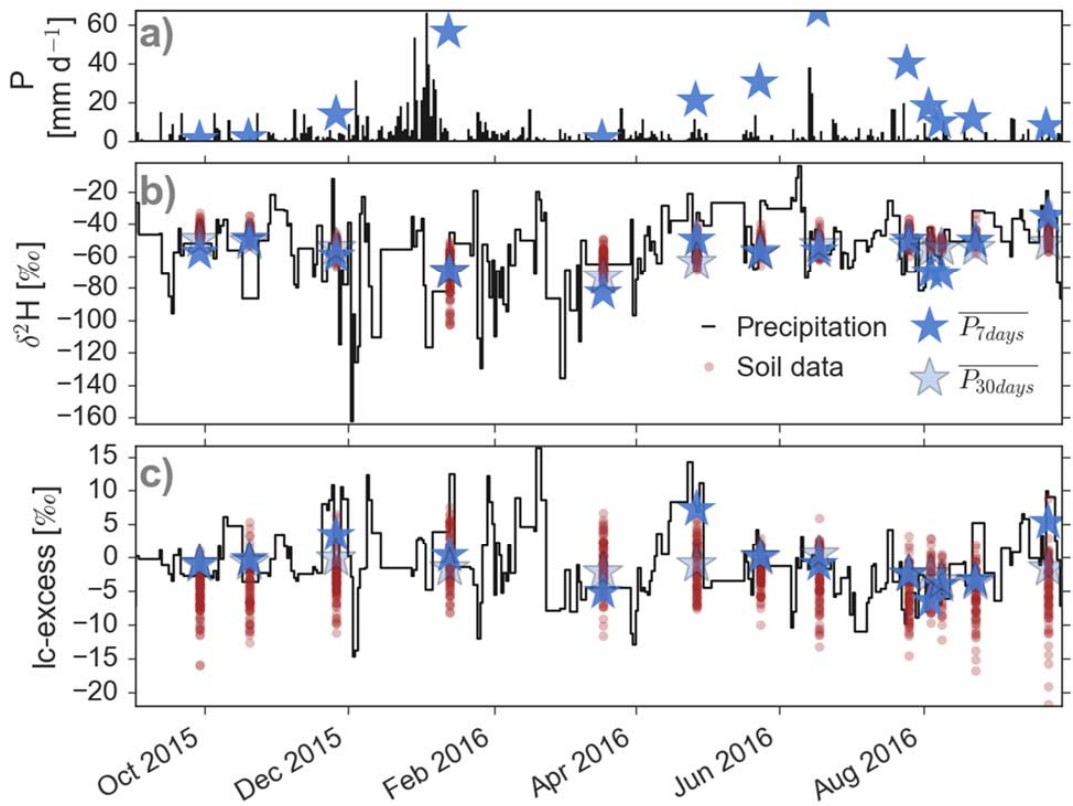

**Figure 2 (a) Daily precipitation sums shown in a bar plot and precipitation sums over the 7 days prior to the soil sampling P₇ as a blue star. (b) Dynamics of the $\delta^2$H and (c) lc-excess in the precipitation input (black line) and soil waters (brown dots). Stars indicate values as weighted averages over 7 days (P₇, blue) and 30 days (P₃₀, light blue) before the day of soil sampling.**





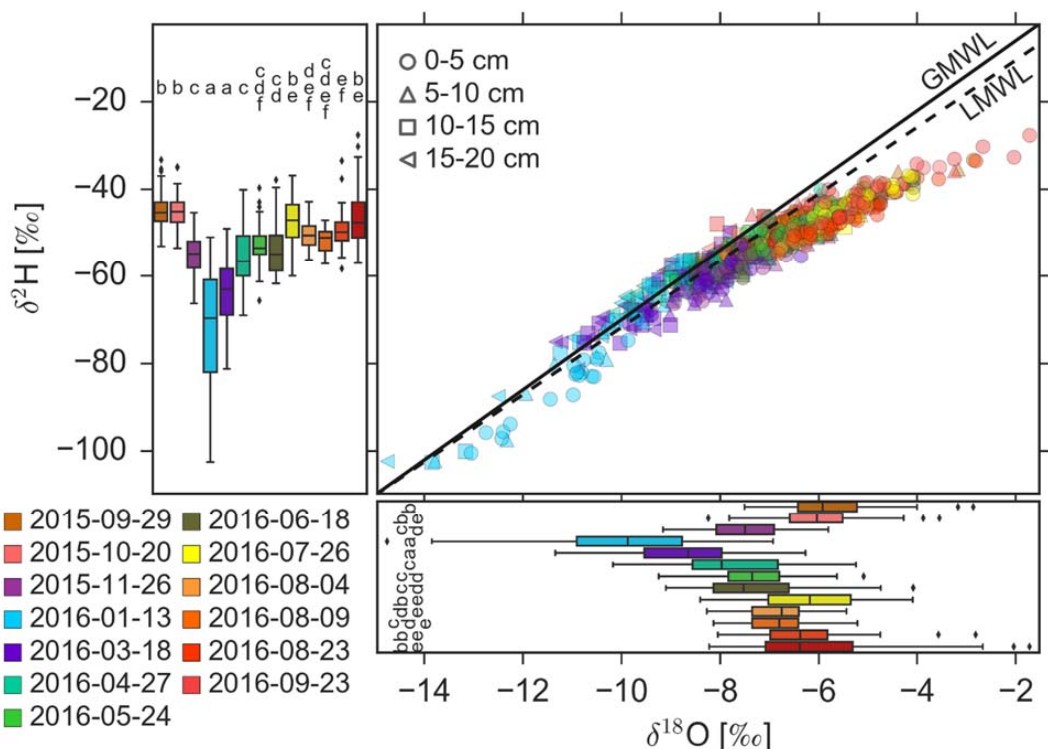

**Figure 3 Dual isotope for all soil sampling campaigns and boxplots for $\delta^{18}O$ and $\delta^2H$ bulked over all four sampling sites. The date of each soil sampling is indicated by colours. The depth of the soil samples is shown by different symbols in the dual isotope plot. Sampling dates which do not have significantly different isotope data according to the post-hoc Dunn test are indicated by the same letter next to the boxplots. For example, the letter "a" at the box plots from sampling day 13 January 2016 and 18 March 2016, indicate that the soil water isotopes do not differ significantly from each other, but to all other sampling days. The global meteoric water line (GMWL, $\delta^2H = 8 \times \delta^{18}O + 10$) is shown with a solid line and local meteoric water line (LMWL, $\delta^2H = 7.6 \times \delta^{18}O + 4.7$) is plotted with a dashed line.**




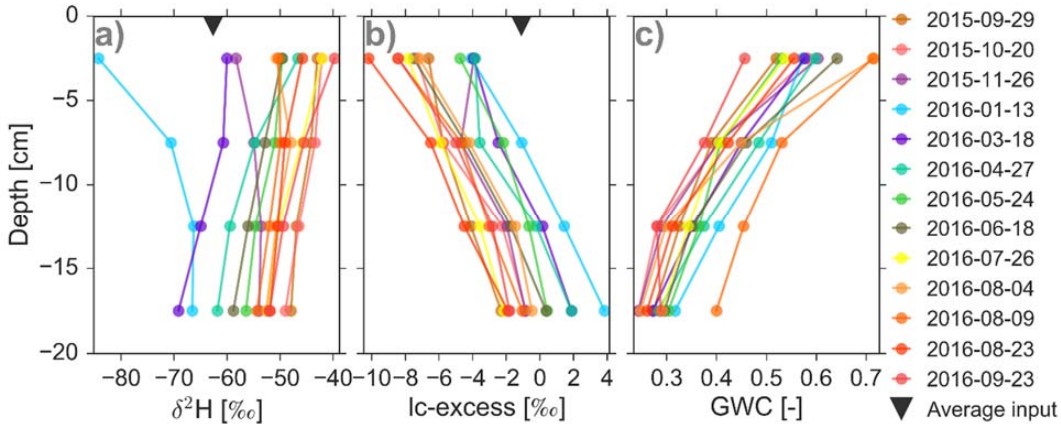

**Figure 4** Depth profiles of (a) $\delta^2$H, (b) lc-excess, and (c) gravimetric water content (GWC) for each sampling day pooled across all sites. Color code indicates sampling days and the black triangle represents the weighted average precipitation input signal during the sampling period ($\delta^2$H = -62.6 ‰ and lc-excess = -1.1 ‰).

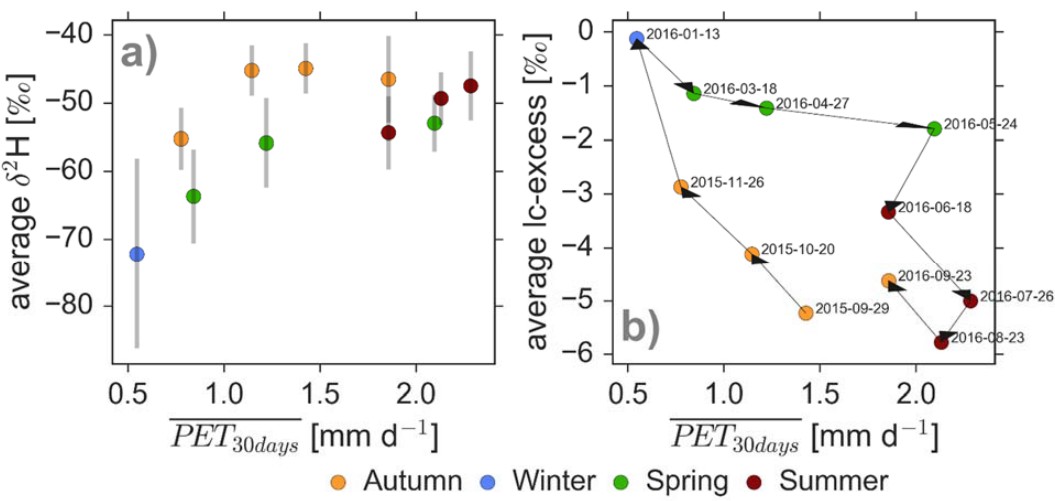

**Figure 5** Variation of the (a) $\delta^2$H and (b) lc-excess in the top 20 cm of the soil as a function of the potential evaporation averaged over the 30 days prior to the sampling day (PET30). The colours of the scatter points indicate the season of the sampling. The error bars in (a) indicate the standard deviation of $\delta^2$H in the soil waters. The arrows in (b) indicate the order of sampling and sampling dates are given.





### 3.2 Spatial soil water isotopes patterns

#### 3.2.1 Differences between study sites

A comparison of the SW $\delta^2$H and $\delta^{18}$O values within the upper 20 cm at the sites for different sampling showed no significant differences between the sites (not shown). Differences of the SW isotope values between sites were limited to the upper 10 cm, with a significantly more depleted $\delta^2$H signal for soil water at SH compared to NF (at 0-5 cm) and SF (at 0-10 cm), when looking at values bulked over the entire sampling period (Figure 6a). For lc-excess, on average soil water samples at NF and SF showed lower values than NH and SH and differences were significant between NF and SH at 0-5 cm depth and for NF and SF compared to NH at 5-10 cm depth (Figure 6b). With regard to GWC (Figure 6c), SH had significantly higher values than the other sites and NF had significantly lower values than the other sites in the upper 15 cm. The median GWCs at NH and SF were very similar, but variability was higher in the latter. The GWC was significantly linearly correlated to the organic content in the soil (Table 1) at all sites and this relationship explains partly the significant higher GWC at SH. As already shown in Table 1, GWC is influenced by the organic matter. This influence was evident during each sampling campaign: the sites with higher LOI tended to have higher average GWCs for the 11 sampling days. The coefficients of determination for the relationship between average LOI and average GWC on the sampling day was generally >0.56 (blue points in Figure 7). Note that these coefficients of determination are limited to a sampling number of 4, given by four sampling sites, while the correlations given in Table 1 are based on n = 20 for each site. This relationship between GWC and LOI persists independently of how many dry days occurred prior to the soil sampling (Figure 7).

While there was no effect of organic matter content on $\delta^2$H and $\delta^{18}$O values, average LOI at the sites showed a relationship with lc-excess on several sampling dates. These sampling campaigns, when the $r^2$ for the relationship between lc-excess and LOI ranged from 0.42 to 0.80, were all characterized by having at least 13 dry days during the 30 day period prior to the sampling. For sampling campaigns with more than 13 dry days over the 30 days prior to the sampling, there was no relationship between lc-excess and LOI ($r^2$ ranged between 0.00 and 0.29, Figure 7). Therefore, there is a significant correlation between the correlation coefficient between lc-excess and LOI and the number of dry days prior to the sampling (r = -0.84, p < 0.01).

The SW lc-excess in the upper 20 cm was usually not statistically different between the sites, since they generally follow a similar pattern (bars in first four columns in Figure 8). At all four sites, the SW lc-excess averaged over the upper 20 cm was significantly < 0 ‰ between May and October and not significantly < 0 ‰ in January. However, the SW lc-excess for NF and SF was often lower than for NH and SH and the lc-excess for NF and SF was significantly < -5 ‰ for some sampling campaigns, while NH and SH was usually not significantly < -3 ‰ (background colour in first four columns in Figure 8).

Further, SH, the site with the highest GWC, had in 6 out of 11 sampling campaigns the lowest slope of EL, while NF, the site with the lowest GWC, had 6 times the highest slope of the EL (Figure S 2).





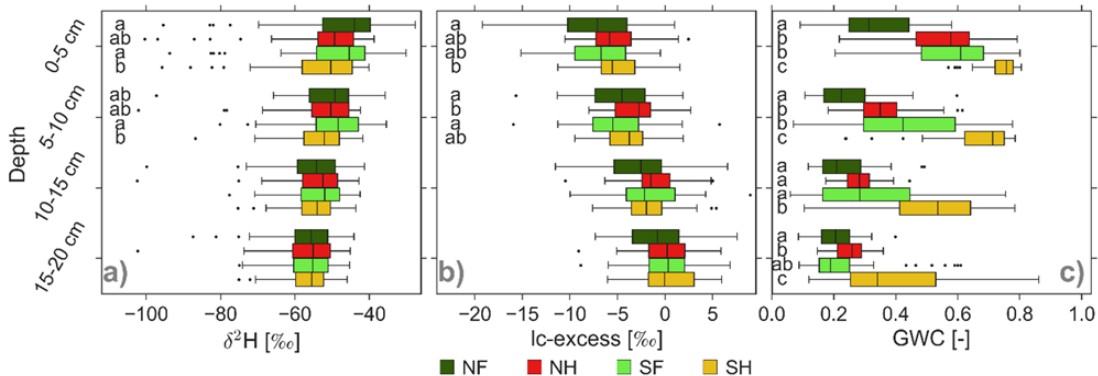


**Figure 6 Site specific (a) δ²H, (b) lc-excess, and (c) gravimetric water content (GWC) for the soils at the four study sites summarized for each site over all sampling dates as function of depth. For depths, where letters are shown, different letters indicate significant differences between the sites for the specific depth.**


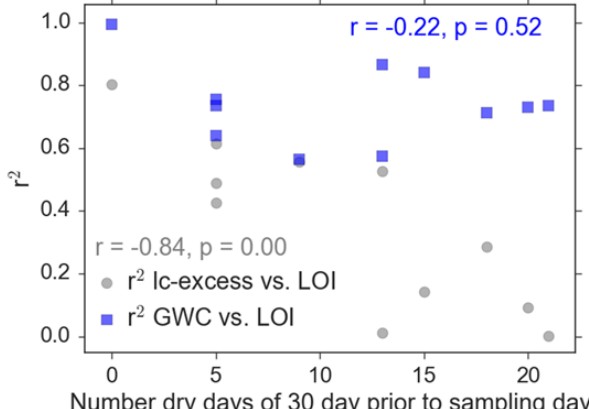


**Figure 7 Relationship between coefficient of determination of the relationship between lc-excess and LOI (grey) and GWC and LOI (blue) with the number of dry days during the 30 day period prior to the soil sampling.**




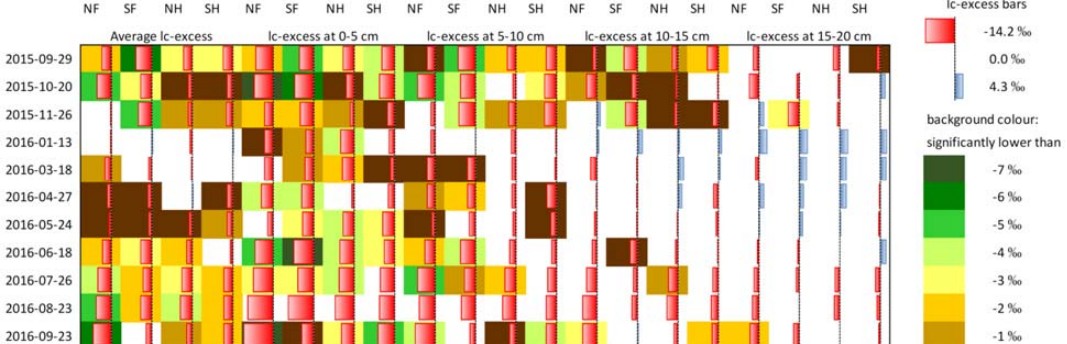


**Figure 8** Average lc-excess values for each study site and each sampling campaign (first four columns), averaged over the 20 cm depth profiles and separated into the 5 cm sampling intervals. The length of the bars represents the lc-excess value (red bars indicate negative lc-excess and blue bars indicate positive lc-excess values). The color of the background provides information if the values are significantly lower than -7 to 0 ‰ at the 95 % confidence interval. White background shows that these samples do not deviate significantly from 0 ‰ or the lc-excess is on average > 0 ‰.

### 3.2.2    Differences with soil depth

During most sampling dates $\delta^2H$ values became more depleted with increasing soil depth (Figure 9a; SW $\delta^{18}O$ very similar and therefore not shown). The $\delta^2H$ values for the top 5 cm were always significantly different compared to the values at 15-20 cm. The $\delta^2H$ values at 0-10 cm were for most sampling campaigns significantly enriched compared to the soil water at 15-20 cm. However, during the November and January sampling, the upper 0-5 cm were more depleted due to the depleted precipitation $\delta^2H$ input (Figure 1b). The highest variability of SW $\delta^2H$ within the sampling depths was found for the samples taken in January and March. Importantly, the variability of SW $\delta^2H$ generally decreased with soil depth.

The lc-excess for the upper 5 cm was always significantly more negative than at 15-20 cm (Figure 9b). In addition, the soil signature at 5-10 cm was significantly more negative than at 15-20 cm; exceptions were November 2015 at 0-5 cm and September 2016 at 5-10 cm. The lc-excess depth profiles had a persistent pattern of steadily decreasing lc-excess values with depth, approaching SW lc-excess of 0 ‰ at 15 – 20 cm depth. The SW lc-excess variability usually decreased with depth, but not for the November sampling (Figure 9b).

The soil water at 15 – 20 cm was usually not significantly < 0 ‰ over the entire sampling period (see also Figure 8). At 10 – 15 cm depth, the SW lc-excess was significantly lower than at least 0 ‰ only between September and November 2015. There were more sampling dates, when the SW lc-excess at 5 – 10 cm was significantly lower than -2 ‰ and usually < 0 ‰ (except for January and August 2016). For the top 5 cm, the lc-excess was partly significantly < -7 ‰ and at least < 0 ‰ throughout the year; with only few exceptions (Figure 8).





The upper 5 cm soil had always significantly higher GWCs than the soil between 15 and 20 cm depth. This pattern of
decreasing soil moisture with depth was persistent over time. The range of the GWC was always much lower below
15 cm than above (Figure 9).

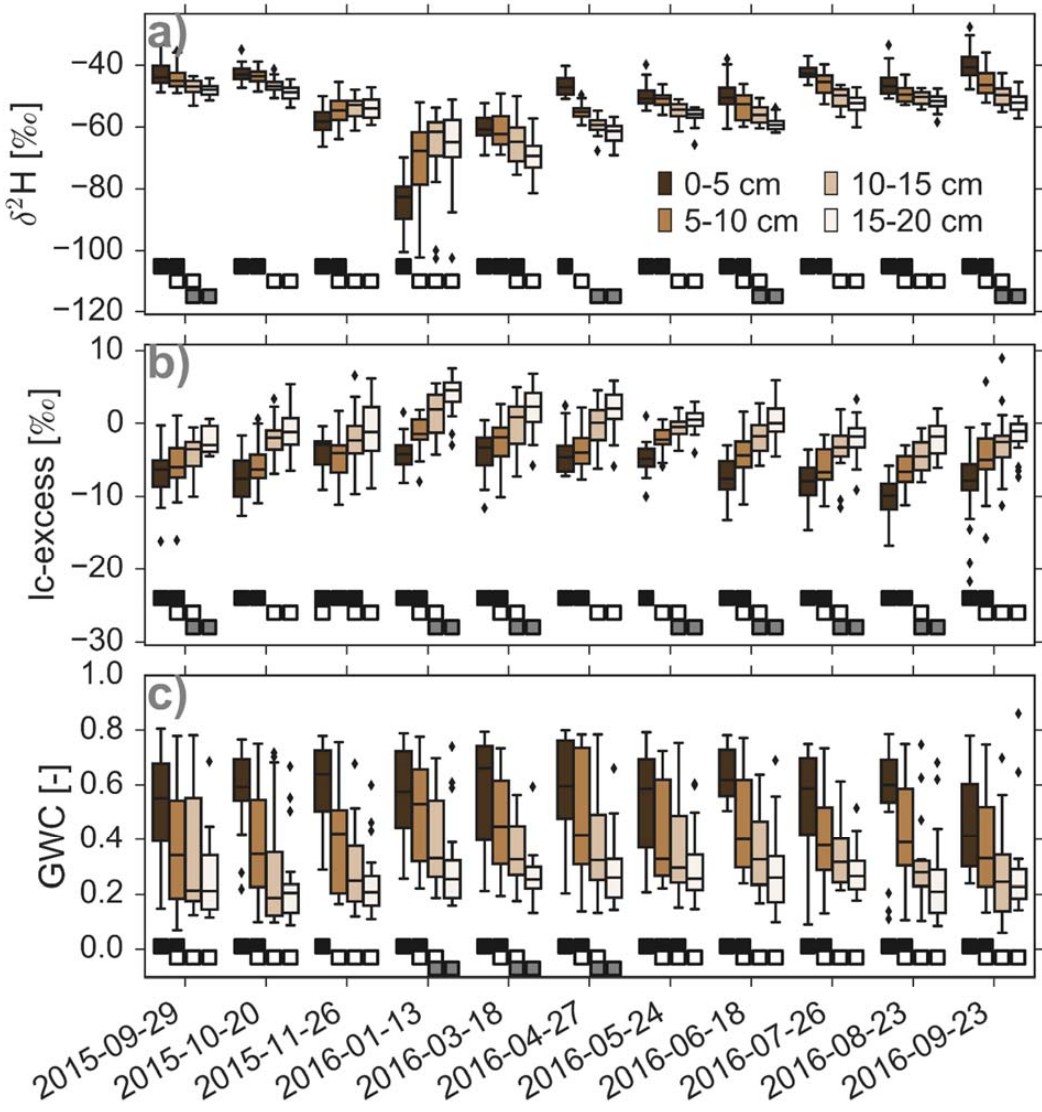


**Figure 9 Differences of (a) δ²H, (b) lc-excess, and (c) gravimetric water content (GWC) over depth for each sampling day. Black, white and grey squares below the boxplots indicate significant differences between the sites for each sampling day according to the Dunn test (i.e. same colour reflects similarities; different colours differences). E.g., for 2015-09-29, samples from 0-5 cm depth were significantly different from samples from the depth 10-15 cm and 15-20 cm, but not significantly different to samples from 5-10 cm depth.**





### 3.2.3 Differences due to vegetation cover

Comparing all soil samples bulked over depth and sampling campaigns, SW beneath Scots pine was significantly more enriched in $\delta^2$H (Mann-Whitney-Wilcoxon Test, p=0.038) and $\delta^{18}$O (Mann-Whitney-Wilcoxon Test, p=0.0062) than the SW beneath heather sites. In addition, lc-excess was significantly lower for the SW at the forested sites compared to the heather sites (Mann-Whitney-Wilcoxon Test, p< 0.01).

The temporal dynamics of the differences of $\delta^2$H (and also SW $\delta^{18}$O, but not shown) values for SW beneath the different vegetation sites show that the SW beneath Scots pine was (except for the sampling in March for $\delta^2$H) always more isotopically enriched than the soil water beneath heather (Figure 10a). However, the differences were only significantly different for four out of the eleven sampling campaigns (i.e., Sept. 2015, Nov. 2015, Aug. 2016, Sept. 2016). SW $\delta^2$H values showed usually higher variability for soils beneath Scots pine for the sampling campaigns between May and October, but during Winter, the variability is generally high for soils beneath both vegetation types.

The SW lc-excess in the upper 20 cm was - except for the January and May sampling campaigns - more negative in the soils beneath Scots pine compared to soils beneath heather (Figure 10b). This difference was significant for more than half of the sampling campaigns. The GWC was also always lower in the soils beneath Scots pines. These differences were most pronounced at the end of the summer and during autumn (Figure 10c).

The differences between the SW isotopic composition under the two different vegetation types mainly stemmed from differences in the shallow soils. Despite being classed as similar podzolic soils, the SWs beneath Scots pine were significantly more enriched in the upper 10 cm in $\delta^2$H (and also $\delta^{18}$O, not shown) than the SW beneath heather (Figure 11a). SW beneath Scots pine had a significantly more negative lc-excess signal for the upper 15 cm soils. Below 15 cm, the differences between SW under Scots pine and SW under heather were not significant (Figure 11b). The GWC of soil beneath Scots pine were over the entire soil profile significantly lower than for the soils beneath heather (Figure 11b).

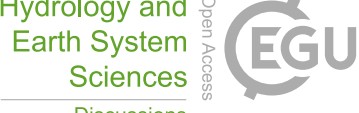



**Figure 10** Differences of (a) $\delta^2$H, (b) lc-excess, and (c) gravimetric water content (GWC) for soils under Scots pine (green) and soils under heather (violet) during one year. Significant differences estimated with the Mann-Whitney-Wilcoxon test between soils under Scots pine and soils under heather for a particular day are indicated by a star.




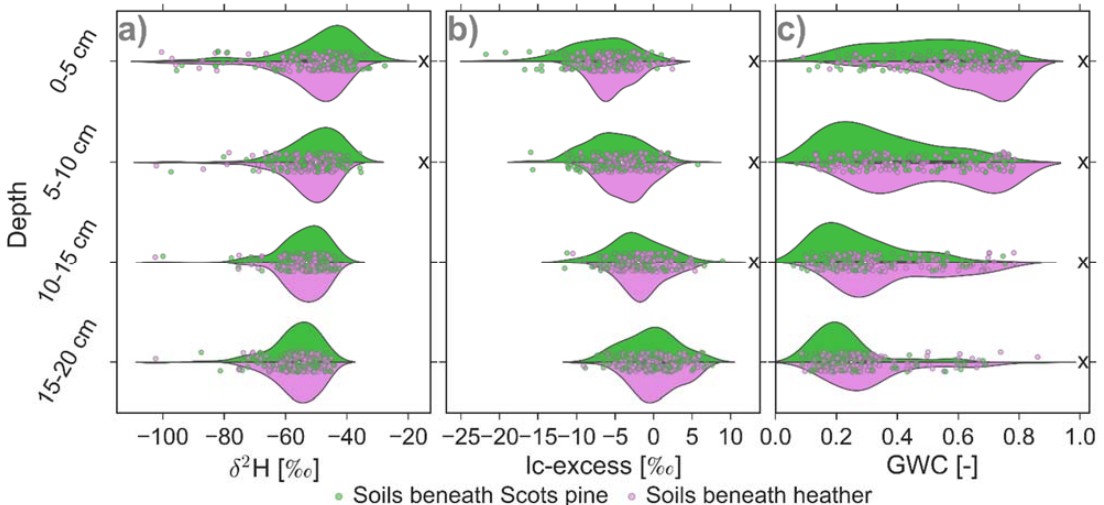

Figure 11 Violin plots showing the distribution as a kernel density estimation of the soil water (a) $\delta^2$H, (b) lc-excess, and (c) gravimetric water content over all sampling campaigns with differentiation between soils beneath Scots pine (green) and soils beneath heather (purple). The symbol "X" at the depths indicates significant differences between the soils under Scots pine and soils under heather for the particular depth as estimated with the Mann-Whitney-Wilcoxon test. Dots indicate the individual sample points.

### 3.2.4 Differences due to aspect

No significant differences were found when splitting the samples according to their aspect. There were neither differences when looking at all sampling depths, nor for any individual sampling depth (not shown). A comparison of the more intensive spatially distributed sampling focusing on the top 10 cm of heather soils in May 2016 at the north facing sites (n = 26) and south facing sites (n = 33) showed also no significant differences between aspects with regard to isotopes and soil moisture. Increasing the sampling numbers for the heather sites when the two additional sampling campaigns in August were included, also did not result in significant differences between the two studied slopes. The median SW $\delta^2$H values for the samples taken at the south and north facing slopes were -50.6 ‰ and -50.9 ‰, respectively. The median SW lc-excess was -3.4 ‰ and -3.3 ‰ for south and north facing slope, respectively. The median GWC was 0.62 and 0.54 for the samples at the south and north facing slope, respectively.

### 3.3 Evaporation estimates

Our findings clearly showed an influence of the vegetation on the evaporation signal in the soil water between April and October. Therefore, for that period we applied the *Hydrocalculator*, as described in the methods section, to estimate the evaporative water losses *f* and evaporation to input ratios (*E/I*), for the soil water beneath Scots pine and heather. For the heather sites, the median (± SD) values of $f_{d2H}$ and $f_{d18O}$ were 4.5±1.0 % and for the forested sites, the values were $f_{d18O}$ = 7.7±1.7 % and $f_{d2H}$ = 8.4±2.9 %. This indicates that between April and October, usually 5 % and 8 % of the soil water was evaporated beneath heather and forests, respectively.





This translates to median values for the ratio between the evaporation and the infiltration (net precipitation) of $E/I_{d2H}$
$= E/I_{d18O} = 5\pm2.4$ % for heather and $E/I_{d2H} = 10.4\pm3.8$ % and $E/I_{d18O} = 9.3\pm2.0$ % for Scots pine during the growing
season. Thus, of 20 mm throughfall and stemflow reaching the soil, 1 and 2 mm will evaporate from the soils beneath
heather and Scots pine, respectively. The residual water will be taken up by the roots or percolate into the subsoil.
**4. Discussion**
**4.1 Mixing of precipitation input and the critical zone water storage**
Our uniquely detailed data set from 11 sampling campaigns over one year revealed the isotopic response of soil water
to variable precipitation inputs. An understanding of the isotopic variability of the precipitation input signal is crucial
for the interpretation of the soil water isotopes in terms of soil evaporation dynamics at the soil-plant-atmosphere
interface. Therefore, we will first discuss how mixing of the infiltrating precipitation within the topsoil provides the
basis for the isotopic enrichment in the soil due to evaporative losses.
Given that the soil at the four sites consists of similar texture, it is not too surprising that the SW generally responded
similarly to the seasonal P $\delta^2H$ and $\delta^{18}O$ input signal. However, we would expect differences in the soil physical
properties due to the differences in the organic matter contents. Our data showed that higher organic matter resulted
in higher GWCs at the sites. That means that the soil water storage is higher for the sites with higher organic material,
which is in line with several other studies (e.g., Hudson, 1994).
While this relationship between GWC and LOI persisted independently of how many dry days occurred prior to the
soil sampling, LOI showed a relationship with lc-excess depending on the dryness. This suggests that during dry
periods, the external atmospheric drivers such as evaporative demand may have a high influence on the SW lc-excess.
In contrast, during wet periods, the soil water storage capacities – here, mainly controlled by organic material –– gain
importance. With increasing new – usually unfractionated – precipitation input, the sites with higher organic matter
content have a different SW lc-excess than the sites with lower organic matter content, because there is a higher mixing
volume for the sites with the high LOI compared to the soils with lower LOI (Figure 7). Hence, future and current
changes in the organic matter content of soils with high organic matter due to, for example, land use or climate changes
(Foley et al., 2005; Rees et al., 2011), would have an impact on the mixing of event and pre-event water in the podzols.
Interestingly, the highest variability of the $\delta^2H$ values in the soil water was found for the sampling after the intense
rainfalls in January 2016 (Figure 5a, Figure 9a), which indicates that the unsaturated zone was not homogeneously
wetted by the exceptionally high precipitation input. The GWC corroborates this, because its variability for the
sampling campaign in January was not lower than for other sampling days (Figure 9b). This suggests that bypass flow
occurred and the stored soil water was not necessarily well-mixed. However, the significantly lower $\delta^2H$ values in the
top 5 cm indicate that, despite the potential occurrence of preferential flow, most of the depleted precipitation input
was stored in the very top soil and the more enriched soil water from autumn was replaced by the event water. The
sampling in January was further special, because the GWC was about as high in the soils beneath Scots pine than in



the soils beneath heather. Hence, differences in GWC between soils beneath the two vegetation types during the rest
of the year as discussed below would stem from different transpiration and soil evaporation rates and not from
differences in the soil water storage capacities.
The higher variability of the SW $\delta^2$H values beneath Scots pine compared to the SW beneath heather (Figure 11)
indicates that flow paths are generally more variable in the forest soils. This is probably due to preferential flows via
larger macropores formed by tree roots that also reach deeper than the shallower roots of heather. While the high
variability of SW $\delta^2$H values indicate preferential flow paths, there was also a clear signal that much of the input water
percolates through the pore matrix, since we see that the depleted winter precipitation signal is still evident at -20 cm
soil depth during the March sampling campaign (Figure 9a).
The relatively high soil water storage volumes in the podzols result in a damping of the $\delta^2$H signal in the SW compared
to the precipitation input. There was also a more damped isotopic signal with soil depth due to increased mixing and
cumulatively larger soil water storage. Because of this damping effect, the differences between the sites (Figure 6a,
Figure 11a) and sampling times (Figure 4a) decreased with depth.
Our data support the findings of Geris et al. (2015b), who showed - with soil water samples extracted with suction
lysimeters - a more damped signal in deeper layers of the soils beneath heather than beneath Scots pine. However, we
did not see a more delayed response beneath Scots pine compared to podzols beneath heather as observed by Geris et
al. (2015b). This discrepancy could arise from the fortnightly sampling frequency by Geris et al. (2015b) and monthly
sampling frequency in the current study, as well the effect of an unusually dry summer in former study.
In contrast to Geris et al. (2015b), who sampled the soil-vegetation units with duplicates, our sampling design with
five replicates allowed to a clearer assessment of the spatial heterogeneity of the subsurface. Our results emphasize
the importance of the need to account for the spatial variability of the pore water isotopic signal. For example, the
variability of the SW $\delta^2$H and $\delta^{18}$O values at -5 cm depth for each site was always higher than the measurement
accuracy of 1.13 ‰ and 0.31 ‰, respectively. This variability of soil water isotope signals will also affect the
interpretation of potential sources of root water uptake (for examples with replicates see McCutcheon et al., 2016 and
Goldsmith et al., 2012). Further, the high variability will potentially impact the application of soil water isotopes for
the calibration of soil physical models and the resulting interpretation (Sprenger et al., 2015b).
The observed mixing processes have implications for modelling of water movement within the critical zone. The  soil
water isotope response seems to follow a replacement of the pore waters by a mixture of newly infiltrated precipitation
and older water, which could probably be described by the advection dispersion equation as frequently applied for
isotope modelling in the unsaturated zone (Adomako et al., 2010; Stumpp et al., 2012; Mueller et al., 2014; Sprenger
et al., 2015b). However, during intense rainfall, the precipitation input seems to partly bypass the matrix leading to
rapid changes in the tracer signal in the subsoil and a generally high variability of the soil water isotopes within the
entire soil profile. Such preferential flow processes might be better represented by soil physical models that take
different mobility of water within the pore space into account, like the dual-permeability (Gerke and van Genuchten,
1993) or dual-porosity (van Genuchten and Wierenga, 1976) approaches. For example, Gouet-Kaplan et al. (2012)





showed with column experiments that a significant amount of pre-event (or "old") water stayed in a sandy pore system.
They found that these processes could be described by a dual-porosity (mobile-immobile) model. Such incomplete
mixing within the unsaturated zone could have an impact on catchment transit time estimates (van der Velde et al.,
2015). Given the high storage volumes of the periglacial deposits (Soulsby et al., 2016b) and the high water mixing
volume in the riparian zone (Tetzlaff et al., 2014) in the Bruntland Burn, it is yet unclear, which role the relatively
slow moving and partly mixed soil waters in the soil matrix at the hillslopes could play regarding long tails of transit
times in the Scottish Highlands, as reported by Kirchner et al. (2010).

### 4.2 Evaporation dynamics within the soil-plant-atmosphere interface

The correlation between the seasonally variable P and SW isotopic signals inhibited an assessment of soil evaporation
processes from either $\delta^2H$ or $\delta^{18}O$. Instead, the SW lc-excess values were independent from the P input and were
therefore indicative of kinetic fractionation due to soil evaporation. The monthly soil water sampling revealed kinetic
fractionation dynamics (in terms of lc-excess) during times of highest potential evaporation for the Scottish Highlands.
The evaporation fluxes were not expected to be high given the relatively low energy environment at the study site.
The soil waters showed, nevertheless, a clear fractionation signal in terms of their lc-excess (25th percentile < -6 ‰
between June and October) and the evaporation line (3.6 and 4.5 between June and October, Table 2). This
fractionation signal in the soil water was of the same magnitude as for surface waters in the peatland drainage network
of a raised bog in the Bruntland Burn, where the lc-excess reached values < -5 ‰ and the EL slope ranged from 3.9
to 4.9 between May and September (Sprenger et al., 2017b). In comparison to arid and Mediterranean environments,
where SW lc-excess can fall below -20 ‰ (McCutcheon et al. (2016) and reviewed in Sprenger et al., (2016b), the
SW lc-excess in the Scottish Highlands remained relatively high. While the lc-excess was usually not significantly
different from zero at 15 – 20 cm soil depth in the studied podzols of the study site, soils studied by McCutcheon et
al. (2016) in a much drier environment (Dry Creek, Idaho) showed that the SW lc-excess from the surface down to -
70 cm was significantly lower than zero.
The high soil water storage contributes to the SW lc-excess dynamics shown in the hysteresis pattern for the
relationship between SW lc-excess and $PET_{30}$, which revealed that there was a delayed response of the SW lc-excess
to the onset and offset of soil evaporation in spring and autumn, respectively. We explain the hysteresis by generally
low soil evaporation fluxes, relatively high humidity in the Scottish Highlands, and high soil water storage. The high
soil water storage means that relatively high evaporation losses are needed in spring to change the $^2H$ to $^{18}O$ ratios by
kinetic fractionation. The variability of SW lc-excess values from July to September was relatively small, indicating
that a steady state between soil water fractionation by soil evaporation and input of unfractionated precipitation was
reached. In autumn, when the soil evaporation ceased, relatively high unfractionated precipitation input was needed
in order to dilute the evaporation fractionation signal of the soil water again (Figure 5).
In contrast to our findings of a pronounced evaporation fractionation in the soil water, Geris et al. (2015b), who
sampled the mobile soil water for its isotopic composition with suction lysimeters, saw little to no fractionation. The
sampling by Geris et al. (2015b) took place at the same/similar sites (podzols beneath Scots pine and heather within





the Bruntland Burn), but during different years. However, the sampling period of Geris et al. (2015b) covered summer
2013, which was exceptional dry (~10 year return period) and would have been therefore very likely to induce
evaporation fractionation in soil waters. We conclude that the differences between the two studies most likely relate
to the different sampling methods. Suction lysimeters, as applied by Geris et al. (2015b), are known to be limited to
sample the mobile water within the pore space. The direct equilibration method applied in our study also samples
more tightly bound waters (Sprenger et al., 2015a). Consequently, we can infer that the mobile waters sampled with
suction lysimeters will be relatively young waters that percolated without experiencing pronounced water losses due
to soil evaporation. In contrast, the soil water data presented in our study, represent a bulk pore water sample, where
more tightly bound waters will be integrating older ages and therefore, affected by kinetic fractionation during periods
of atmospheric evaporative demand. However, comparisons between mobile and bulk soil water isotopic signals
showed that tightly bound water at soil depth deeper than 10 cm was usually more depleted than the mobile waters
during spring or summer, due to filling of fine pores of a relatively dry soil with depleted precipitation several months
earlier (Brooks et al., 2010; Geris et al., 2015a; Oerter and Bowen, 2017). Our study shows that, in contrast to old
isotopically distinct infiltration water, also the legacy of evaporation losses over previous months allows for separating
between pools of different water mobility in the soil-plant-atmosphere interface.
So far, mostly cryogenic extraction was used to analyse the isotopic composition of bulk (mobile and tightly bound)
soil waters and relate these data to for example root water uptake pattern. However, recent experimental studies
indicated potential for fractionation during the cryogenic extraction (Orlowski et al., 2016), probably induced by
water-mineral interactions (Oerter et al., 2014; Gaj et al., 2016). The direct-equilibration method – as a potential
alternative analysis method - we applied in our study was so far usually limited to study percolation processes (e.g.,
Garvelmann et al., 2012; Mueller et al., 2014; Sprenger et al., 2016a). To our knowledge, only Bertrand et al. (2012)
used the direct-equilibration method to study the water use by plant water use and the influence of evaporation
fractionation, but they bulked the soil data of the upper 20 cm.
However, our results emphasize the importance of sampling the upper most soil layer when studying plant-water
interactions and soil evaporation dynamics. The evaporation was predominantly taking place at the interface to the
atmosphere, where the lowest SW lc-excess values were found throughout the sampling period. Due to the relatively
wet conditions of the studied soils, the evaporation signal dropped sharply within the first 15 cm soil depth and was
always significantly lower at the top 5 cm compared to the SW at 15 – 20 cm depth. Geris et al. (2015a) sampled with
cryogenic extraction the isotopic composition of the bulk soil water at -10 cm soil depth at the site NF during the
summer of 2013. They found soil water isotopes of the same magnitude at -10 cm ($\delta^2$H between -40 and -55 ‰), but
their sample size was too little to assess evaporation fractionation patterns.
The pronounced differences of the soil water samples in the dual isotope space over little soil depth is crucial when
assessing potential water sources of the vegetation. The lc-excess signal could therefore provide additional information
as an end-member when applying mixing models to derive root water uptake depths and times, since $\delta^2$H and lc-excess
do not necessarily correlate. So far, usually either $\delta^2$H or $\delta^{18}$O, but not lc-excess are being considered to delineate
water sources of the vegetation (Rothfuss and Javaux, 2016). The relatively high dynamic of the isotopic signal during




the transition between the dormant and growing seasons underlines the importance of not limiting the soil water
isotope sampling to few sampling campaigns (usually n ≤ 3 in published literature), when investigating root water
uptake patterns. In the light of recent studies dealing with potential water sources of vegetation that showed how
variable the plant water isotopic signal can be over a year (Hervé-Fernández et al., 2016; McCutcheon et al., 2016), a
proper understanding of the temporal variability of the potential water sources (i.e., bulk soil water, groundwater,
stream water) appears to be critical.

### 4.3 Vegetation affects critical zone evaporation losses

The vegetation was the main reason for differences in the soil water evaporation signal between the four study sites.
While we limited the study to one soil type, we still saw differences between the soils in terms of their organic matter
content leading to different soil water storage capacity. However, these differences seemed to only influence the
evaporation signal during periods of high wetness, when evaporation is already likely to be low (Figure 7). The aspect
and exposition of the slopes did not affect the soil water evaporation signal, which probably can be explained by the
low angle for the north- and south-facing slopes at our site. Differences in net radiation for the two studied slopes
were found to be small with the north facing slope receiving about 4 % less radiation than the south facing slope
between April and October (Ala-Aho et al., in review).
The significant differences of the SW lc-excess in soils beneath Scots pine and heather are due mainly to the
differences of the vegetation structure. The heather forms a dense soil cover just about 20 cm above the ground with
an understory of mosses and lichens. Thus, the vegetation shades most of the soil and generates a microclimate with
little direct exchange between soil vapour and atmospheric vapour. The soil climatic conditions beneath heather are
therefore characterized by higher relative humidity and less radiation input than for the soils beneath Scots pine. In
addition, the soils beneath the Scots pine are not densely vegetated by understory, which allows for more open
exchange between soil vapour and atmospheric vapour at the forested sites. A similar conclusion of microclimatic
conditions driving the differences in the evaporation signal was drawn by Midwood et al. (1998), who also reported a
higher isotopic enrichment of soil surface soil waters under groves and woody clusters than under grassland.
The fractionation signal will generally be more pronounced for evaporation losses from a smaller water pool than a
larger one. Higher canopy storage, higher interception losses and reduced net precipitation in the forested stands,
where throughfall is about 47 % of precipitation, compared to the heather with throughfall being about 35 % of
precipitation (Braun, 2015) will influence the GWC of the soils beneath the two vegetation types. The higher
transpiration rates of the trees (Wang et al., in review) compared to the heather further influences the soil moisture
dynamics, leading to significantly lower GWC in the forested sites. In consequence, soil evaporation taking place
from the drier soils beneath Scots pine will result in higher evaporation fractionation than for soil evaporation from
soils beneath heather.
The evaporation estimates are limited to the period of highest evaporation fluxes in the Bruntland Burn and are likely
to be lower than what we present, since we relate the isotopic enrichment to the long term average input signal in our
calculations. However, the assumption of relating the enrichment to the average input seems to be valid, since we have





seen the steady state balance of unfractionated input and kinetic fractionation due soil evaporation during the summer
months. Nevertheless, detailed estimates of the evaporative fluxes require transient modelling of all water fluxes (i.e.,
precipitation, transpiration, evaporation, recharge) and their isotopic composition, which will be subject of future
work.
The frequently measured soil water isotope data we have presented and the inherent evaporation signal can help to
calibrate or benchmark the representation of soil-vegetation-atmosphere interactions in tracer aided hydrological
modelling from the plot (Rothfuss et al., 2012; Sprenger et al., 2016b) to the catchment scale (Soulsby et al., 2015;
van Huijgevoort et al., 2016). So far, the isotopic composition of mobile water has been used to better constrain semi-
distributed models (Birkel et al., 2014; van Huijgevoort et al., 2016). Including the isotopes of the bulk soil water,
could allow for an improved conceptualization of the soil-plant-atmosphere interface, which is crucial for an adequate
representation of evaporation and transpiration. Especially the marked differences of the evaporation signal within the
first few centimetres of the soil depth will significantly affect the age distribution of transpiration (Sprenger et al.,
2016c), evaporation (Soulsby et al., 2016a) or evapotranspiration (Harman, 2015; Queloz et al., 2015; van Huijgevoort
et al., 2016).
**5. Conclusions**
Our study provides a unique insight into the soil water stable isotope dynamics in podzolic soils under different
vegetation types at a northern latitude site. We showed that, despite a relatively low energy environment in the Scottish
Highlands, the temporal variability of soil water isotopic enrichment was driven by changes of soil evaporation over
the year. The monthly frequency of the soil water isotope sampling corroborates the importance of covering transition
periods in a climate with seasonally variable isotopic precipitation input signals and evaporative output fluxes. Missing
sampling these periods of higher temporal variability in spring and autumn could pose problems when referring plant
water isotopes to potential soil water sources or when using soil water isotopic information for calibrating hydrological
models. Especially the delayed response of the soil water lc-excess to evaporation (hysteresis effect) provides valuable
insight into how unfractionated precipitation input and kinetic fractionation due to soil evaporation both affect mixing
processes in the upper layer of the critical zone. The fact that the evaporation signal generally disappears within the
first 15 cm of the soil profile emphasises the importance and the spatial scale of the processes taking place at the soil-
vegetation-atmosphere interface of the critical zone.
The vegetation type played a significant role for the evaporation losses, with a generally higher soil evaporation signal
in the soil water isotopes beneath Scots pine than beneath heather. Notably, these differences - as indicated in the soil
water lc-excess - remain limited to the top 15 cm of soil. The vegetation cover directly affected the evaporation losses
with soil evaporation being twice as high as beneath Scots pine (10 % of infiltrating water) compared to soils beneath
heather (5 % of the infiltrating water) during the growing season.
The presented soil water isotope data of exceptionally high sampling frequency in time (monthly) and space (5 cm
increments) will allow us to test efficiencies of soil physical models in simulating the water flow and transport in the



critical zone, but provide also a basis to benchmark hydrological models to realize a more realistic representation of
the soil-vegetation interactions when simulating water fluxes and their ages within catchments.


**Appendices**

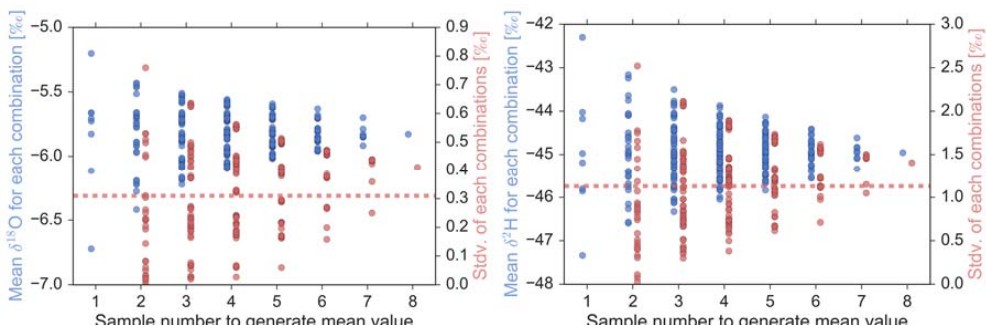


**Figure S 1 Spatial variability of (left) $\delta^{18}$O and (right) $\delta^2$H in soil water and the effects of taking several samples in parallel**
**to get average values (blue dots and primary y-axis) for the samples if only having one sample (1 on the x-axis) or averaging**
**over 2 to 8 samples (2 to 8 on the x-axis). Red dots indicate the standard deviation of the different combinations and the red**
**dotted line shows the precision of the isotope analysis.**








**Figure S 2 Dual isotope plots for each sampling day showing the pore water isotopic composition and the isotopic signal in the precipitation averaged over the 7 days (black stars), 14 days (grey stars), and 30 days (light grey stars) prior to the sampling date. Colors indicate the sampling site and marker indicate the sampling depth. Solid line shows the GMWL and the dotted line represents the local meteoric water line (LWM: $\delta^2H = \delta^{18}O \times 7.6 + 4.7$ ‰).**






**Author contribution**

D.T., C.S. and M.S. established the sampling design together, M.S. conducted the field and laboratory work and statistical analysis, all authors were involved in the data interpretation, M.S. prepared the manuscript with contributions from both co-authors.

**Competing interests**

The authors declare that they have no conflict of interest.

**Data availability**

The data is available upon request.

**Acknowledgements**

We are thankful for the support by Audrey Innes during all laboratory work. We further thank Jonathan Dick for running the isotope analysis of precipitation samples and Annette C. Raffan for her support in the soil texture analysis. We would also like to thank the European Research Council (ERC, project GA 335910 VeWa) for funding.





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





**Table 1 Vegetation and soil profile characteristics of the four study sites: Percentage of coarse gravel (>20 mm diameter)**
**and fine gravel (20-2 mm diameter) of the soil sample, percentage of sand (S, 2-0.6 mm), silt (Si, 0.06-0.002 mm), and clay**
**(C, <0.002 mm) of the fine soil matrix. Percentage of organic content in the soil as loss on ignition (LOI), correlation**
**characteristics for LOI with depth and with gravimetric water content (GWC), and fine root density as percentage of total**
**root mass. Note that for SH, NF, and SF, not enough fine soil was left for soil texture analysis after ignition of organic**
**material at 550 °C and that for NF and SF, no root density could be measured due to root thickness and stone content. ±**
**indicate standard deviations out of five replicates for LOI values and range of two and three replicates for root density,**
**respectively.**

| Site | Vegetation | Depth [cm] | Coarse gravel [% of soil] | Fine gravel | Sand [% of fine soil] | Silt | Clay | LOI [% of soil] | LOI vs. Depth | LOI vs. GWC | Fine root density [% of total roots] |
|------|-----------|------|------|------|------|------|------|------|------|------|------|
| NH | Heather | 0-5 | 0 | 0 | 74 | 22 | 4 | 26 ± 15 | | | 82 ± 7 |
| | | 5-10 | 0 | 5 | 77 | 19 | 4 | 11 ± 5 | r = 0.66; | r = 0.73; | 15 ± 5 |
| | | 10-15 | 0 | 11 | 78 | 18 | 4 | 8 ± 2 | p < 0.01 | p < 0.01 | 3 ± 1 |
| | | 15-20 | 2 | 10 | 80 | 16 | 4 | 9 ± 6 | | | 0 |
| SH | Heather | 0-5 | 0 | 0 | - | - | - | 92 ± 4 | | | 56 ± 18 |
| | | 5-10 | 0 | 1 | 57 | 35 | 8 | 79 ± 14 | r = 0.72; | r = 0.87; | 24 ± 9 |
| | | 10-15 | 0 | 5 | 78 | 18 | 4 | 54 ± 27 | p < 0.01 | p < 0.01 | 16 ± 6 |
| | | 15-20 | 0 | 18 | 81 | 16 | 3 | 50 ± 39 | | | 4 ± 2 |
| NF | Scots pine | 0-5 | 0 | 1 | - | - | - | 28 ± 11 | | | |
| | | 5-10 | 0 | 2 | 80 | 17 | 3 | 31 ± 20 | r = 0.42; | r = 0.70; | |
| | | 10-15 | 0 | 0 | 77 | 18 | 5 | 20 ± 18 | p = 0.06 | p < 0.01 | |
| | | 15-20 | 0 | 2 | 77 | 20 | 3 | 14 ± 10 | | | |
| SF | Scots pine | 0-5 | 0 | 1 | - | - | - | 95 ± 1 | | | |
| | | 5-10 | 12 | 27 | 84 | 12 | 4 | 58 ± 36 | r = 0.69; | r = 0.91; | |
| | | 10-15 | 15 | 22 | 85 | 12 | 3 | 40 ± 23 | p < 0.01 | p < 0.01 | |
| | | 15-20 | 18 | 18 | 81 | 16 | 3 | 45 ± 39 | | | |







**Table 2 For each soil sampling campaign: the PET on that sampling day; PET as average over the 30 days prior to the sampling day ($PET_{30}$); lc-excess of the precipitation input weighted averaged over 7 ($P_7$) and 30 ($P_{30}$) days, respectively; precipitation summed over 7 ($P_7$) and 30 ($P_{30}$) days prior to soil sampling; total soil sample number; minimum, median, maximum, and interquartile range (IQR) of the soil water lc-excess data; slope and intercept of the evaporation line (note that regression is significant at the 99 % confidence interval); and mean gravimetric water content (GWC).**

| | PET | | P lc-excess | | P | | Sample | Soil water lc-excess | | | | Evaporation line | | GWC |
| | [mm d$^{-1}$] | | [‰] | | [mm] | | [-] | [‰] | | | | Slope [-] | Intercept [‰] | [-] |
| Date | PET | $PET_{30}$ | $P_7$ | $P_{30}$ | $P_7$ | $P_{30}$ | n | min. | median | max. | IQR | | | mean |
|---|---|---|---|---|---|---|---|---|---|---|---|---|---|---|
| 2015-09-29 | 2.7 | 1.4 | -0.8 | -1 | 1 | 66 | 91 | -16.1 | -5 | 1.1 | 4.2 | 4 | -21.4 | 0.39 |
| 2015-10-20 | 1.2 | 1.1 | -0.2 | -0.7 | 2 | 60 | 78 | -12.6 | -4.1 | 5.3 | 5.2 | 3.6 | -23.3 | 0.37 |
| 2015-11-26 | 1.1 | 0.8 | 3.4 | -0.2 | 14 | 94 | 80 | -11.1 | -3 | 6.5 | 3.9 | 4.6 | -21.1 | 0.37 |
| 2016-01-13 | 0.4 | 0.6 | 0.4 | -1.6 | 56 | 434 | 77 | -8.2 | -0.8 | 7.5 | 6 | 8.2 | 10.8 | 0.42 |
| 2016-03-18 | 0.6 | 0.8 | -5.2 | -2.1 | 2 | 33 | 79 | -11.6 | -1.3 | 6.7 | 5.4 | 5.5 | -15.2 | 0.37 |
| 2016-04-27 | 0.9 | 1.2 | 7.3 | -1 | 21 | 80 | 79 | -7.7 | -1.8 | 5.8 | 6.4 | 5.1 | -16.2 | 0.40 |
| 2016-05-24 | 1.6 | 2.1 | 0.1 | 0.1 | 31 | 66 | 80 | -10 | -1.4 | 2.9 | 3.7 | 4.9 | -17.1 | 0.36 |
| 2016-06-18 | 2.6 | 1.9 | -1.1 | 0.5 | 67 | 105 | 80 | -13.2 | -3.2 | 5.9 | 5.2 | 4.5 | -21.6 | 0.38 |
| 2016-07-26 | 2.2 | 2.3 | -2.5 | -2.5 | 40 | 98 | 72 | -14.6 | -4.5 | 3.2 | 5.1 | 4.6 | -19.1 | 0.37 |
| 2016-08-23 | 1 | 2.1 | -3.4 | -3.6 | 12 | 54 | 64 | -16.7 | -6 | 2 | 5.1 | 3.8 | -25.4 | 0.35 |
| 2016-09-23 | 0.4 | 1.9 | 5.3 | -1.6 | 8 | 51 | 77 | -21.7 | -4.1 | 8.9 | 6.2 | 4.3 | -19.9 | 0.33 |