# Peer review of "Soil water stable isotopes reveal evaporation dynamics at the soilplant-atmosphere interface of the critical zone"

_Hydrology and Earth System Sciences, 2017_

## Referee Comment (RC1) · Anonymous Referee #1 · 21 Mar 2017

Review of "Stable isotopes reveal evaporation dynamics at the soil-plan-atmosphere of the critical zone" by Sprenger and colleagues.

The manuscript describes evaporation dynamics of soil water in podzolic soils in the Scottish Highlands that are then related to surface vegetation and aspect. The authors sampled soils monthly and analzyed the  $\delta$ 18O and  $\delta$ D stable isotope values of soil water from four different depths for one year using an isotopic equilibrium method. The authors inferred evaporative losses from the soil water profile as well as explored relationships between isotope fractionation and deuterium excess (normalized to the local meteoric water line) with soil organic matter characteristics and corresponding vegetation (forest and heather). They hypothesized that their isotopic patterns were

related to precipitation inputs and mixing processes in the soil. They found the unique characteristics of the research site to dominate the isotopic patterns they detected, especially at the near surface. In particular, the high organic matter content of the soil served as an important storage pool that potentially dampened an evaporative signal in the water isotopic patterns. During summer, when evaporative potential is high, the evaporative signal was strongest in the upper 15 cm.

The experimental design is tailored for testing vegetation influence on site hydrology, the methods are appropriate and the analyses were exhaustive. However, a justification or more information is needed regarding the use of hydrocalculator. The issue at hand is that at the core of hydrocalculator is the Craig-Gordon model, which is designed for open water surfaces ÅňÅň– which the soil clearly is not. There are other models that consider diffusion in the unsaturated zone (i.e., Barnes and Allison, Journal of Hydrology, 1988) and they are able to model the profile while estimating an evaporative flux. In the study under consideration, estimating evaporation is not the focus per se, so removing these potentially erroneous estimates wouldn't be a problem. The figures are in general useful, however, I am afraid figure 8 is too busy to extract the point being addressed without investing a significant amount of time.

Despite the issue of the evaporation estimates, it is clear that evaporation occurs within these soils and the role of vegetation is highly relevant. And while this is interesting, it is not entirely unexpected. Emphasis is also placed on the high frequency of the sampling (11 campaigns), but with advent of portable CRDS lasers even monthly samples are considered a rather low frequency sampling strategy (for example, see Volkmann et al., New Phytologist, 2016). Modelling is mentioned in the discussion (L198-213; pg-22-23) although no modelling is performed and the results from the study are not really brought into the modelling discussion directly. There is a long history of investigating and modelling the soil water isotopic profile, and again while the results are interesting for this particular site, the title seems to promise more than the study can deliver. The merit in the study lies in the site-specific nuances such as the role of organic matter in

effecting soil water capacity and the subsequent isotopic mixing that occurs.

Specific comments: Title: It is not exactly clear what is meant by "the soil-plantatmosphere interface of the critical zone". In this study soil water is measured, why not just state this?

Abstract: Page1, L13: Because this paper is not a test of the method, it is necessary to report it here.

Page1, L22-25: I would argue that this sentence can be deleted.

Introduction:

Page1, L30: remove "well" from well understood so that it reads simply "insufficiently understood"

P2 L2: I think it is the age distribution of the water that is used in evapotranspiration that is meant here, and not the age of the flux.

P2 L10: perhaps introduce the term isotopic fractionation here, not all readers will understand the relevance of this process.

P2 L32: This sentence needs to be restructured.

Page 3: Research question 1: Instead of "critical zone", I suggest soil profile.

Research question 2: This has already been done before. Better to restate your question/goal as to estimate evaporation of the site based on the water stable isotope values within the soil profile. This section might need to be removed if the hydrocalculator approach cannot be justified.

Research question 3: This is very vague. What is meant by feedbacks? And, how do you expect them to vary spatially and temporally?

P3 L20: catchment "is" covered

Page 5, lines 21-25: Please briefly explain how the 'equilibrium method' works.

P5 L26: Please explain how the standard water was "sampled"?

P7 L3: Please discuss the "effect of antecedent conditions". What might we expect that occurs during the time window you suggest?

Page 7, Lines 4-6: Could you briefly explain why exactly you used these number of days (7 and 30)? Furthermore, how were the isotopic signals weighted? This whole approach is rather unclear.

Page 7, Line 13: Did you mix the isotopic results from 5 and 10 cm for achieving f?

Page 7, Lines 15-17: I have some concerns about this approach. What is the purpose of applying this correction to  $\delta$ S? This is not fully clear. Secondly, did you correct the  $\delta$ P for the net precipitation (mm), which was different between from month to month? This would affect the average  $\delta$ 18O and  $\delta$ 2H of precipitation.

P8 L16-17: What is the reason for testing for significance along such a small range? Isn't 0 and 1‰ already in the measurement precision limits? What value is relevant to determine significant evaporation?

P8 L21: Pearson not Parson

Figure 3. I really like this figure, especially with the temporal patterns along the secondary plots.

Figure 5 b. The pattern in Ic-excess over the season is interesting but what qualifies this as hysteresis and not just seasonal changes in PET? This plot needs error bars along both axes.

Page 14, Lines 22-23: What is 'organic content'? Maybe "organic matter content" is more appropriate.

Figure 7. Is there a reason why LOI should change in such a short time period? The x label of this figure is really confusing.

P16 L74: I think this is in reference to 0‰ lc-excess

Page 17 L85: Should it be "depths" instead of "sites"?

Discussion:

P21 L159: Be careful, technically even precipitation has undergone fractionation!

P21 L170: I assume the dynamics at 5cm are being described here. Is the replacement of autumn water the only possibility here? Is it not also possible that mixing (which the authors advocate elsewhere) is responsible? Are they mutually exclusive?

P21 L171: "further special" is a bit awkward

P22 L172-174: Time reference is lost here. Is the rest of the year for any time that does not occur in January? It looks like this pattern was also emerging in November of 2015 (probably also established in December).

P22 L190-197: I don't think this part of the discussion warrants a whole paragraph.

Page 22, Lines 192-195: This statement is not fully clear. What is the importance of the accuracy? Please remember that you are only talking about the top 5 cm. So root water uptake will depend on the plant species and rooting depth.

P23 L217-218: "kinetic fractionation dynamics" is a little cumbersome; it may be easier to the reader if you simply refer to evaporation when referencing the isotopic effect.

P24 L246: citation

P24 L255: Do you mean isotopically depleted infiltration water? I don't immediately see how this study shows that the "the legacy of evaporation losses" allows for separating pools of different water mobility in the SPA interface. Can you make this more apparent?

P25 L279: Where is the number 3 coming from? I think a few citations or reviews might help back this up. I think it is also important to keep the context of the research

question in mind when assessing another study's design.

P25 L288: Isn't the storage capacity referred to earlier relevant here as well?

P25 L302-303: citation

P26 L345: I don't think the word "exceptionally" is warranted here, although the study is data rich. This sentence is also structured in a strange manner (i.e., "but also" when there isn't a contrast to begin with). I don't think the final statement is justified. Are modelers calling for higher spatial and temporal isotope data? Is this listed as a research priority in the literature? Do the data presented here help realize a realistic representation of "soil-vegetation" interactions? Statements like these are beyond the scope of this study.

---

## Referee Comment (RC2) · Anonymous Referee #2 · 29 Mar 2017

General comments

The authors studied the influence of vegetation on water fluxes in the upper soil compartment of the Scottish Highlands by means of stable water isotopes. Soil samples were taken eleven times over the course of a year and analyzed for their isotopic composition using the direct equilibration method. The authors nicely visualized their results. However, they should consider cutting down the number of figures. I think the paper length should also be reduced by at least five pages, which would help focus on the most important points. There are many repetitions, which unnecessarily blow up the manuscript. What did the authors really expect to find? I think there could also be a more compelling title that illustrates immediately to the potential reader what exactly

the paper is about. The title should focus more on the actual findings as the manuscript does not present atmospheric data or detailed data on vegetation (e.g. rooting depth and density) anyway. The authors pose three research questions. In my opinion, these questions could be more precise. In particular, the third research question cannot really be answered by the results – especially not the atmospheric component. With regard to the soil samplings, I would not consider the sampling strategy as high frequent, especially against the background of portable laser spectroscopes which can indeed measure water isotopic composition in-situ with high frequency. The authors describe the soil texture of the upper 20 cm as mainly loamy sand. A table, which compiles all soil properties, would be helpful at this point. Soil properties have been shown to affect the extraction method's isotope results. Do the authors have data on the soil mineralogy (clay mineral composition)? The applied direct equilibration has several downsides: It is less precise for more clayey soils and soils with low water content; storage time is also an issue as it can lead to evaporative water loss through the bag (How long were the bags stored prior to analysis in the present study?) Furthermore, soil organic matter content has been proven to have an effect on gained isotope results. The authors should consider these aspects when discussing their data. In sum, I think this paper will make a good addition to the soil water isotope literature, although it does not contain much novel or surprising findings. However, the authors did a great job in data analyses and presentation.

Specific comments

P 1 L 13: $\delta$18O

P 3 L 1: Thus,...

P 4 L 14-29: Described in too much detail; consider compiling important soil data in a Table

P 5 L 3-8: Far too detailed

P 5 L 24: Not necessary to state model and serial number of the isotope analyzer

P 6 L 7: Not necessary to reference the python module; please change throughout the manuscript

P 8 L 4 –P9 L3: This whole section is again too long. Please condense

P 9 L 6: Different font used

P 11 Fig. 2: for a) I would suggest to plot the rainfall amount data inversely (top-down) and either change the scale of the axis or the size of the blue star so that they are not cut off; for b) consider including moving averages through the soil data (e.g., moving average for the top and subsoil); describe the color code of the soil data (light brown dots stand for. . .)

P 13 Fig. 4: This figure does not add much information; consider deleting this figure. Does the average precipitation input signal represent a 1-yr mean?

P 16 chp. 3.2.2.: Include this section in results section 3.1 as it does not add too much new information

P 20 chp. 3.2.4 Delete this section. There are no sig. differences in isotopic signatures when considering the aspect.

P 20 L n134 ff: Repetition; consider deleting

P 21 L 167-68: bypass flow, really; not so much differences over depth here

P 22 L 175 : Is throughfall data available for these sites to underline this statement?

P 24 L 260: Add Gaj et al. (2017b)

P 24 L 265: This is not a new finding and not really surprising.

P 24 L 269: The authors compare their study with results by Geris et al. (2015a) quite frequently. Is the vegetation cover comparable in both studies?

P 24 L 277: Debatable that the authors state to see highly dynamic isotope signals.

P 25 L 290: What exactly is the angle in your case?

P 25 L 293: gramma: . . .are mainly due to. . .

P 26 L 318: In my opinion, the present study does not really unravel interactions occurring in the soil-plant-atmosphere continuum but adds to process understanding of water fluxes through the soil compartment.

————————————————————

---

## Author Comment (AC1) · 22 Apr 2017

**Response to comments by Anonymous Referee #1**

*We thank Referee #1 for reviewing our paper and their positive feedback on the methods and analysis that we present in our manuscript. We are pleased to see that we could convey the message of the presented study in a convincing way, given the synopsis Referee #1 provides at the beginning of their review. We will first comment on the major issues raised by Referee #1 and further respond to each specific comment by Referee #1 below.*

The manuscript describes evaporation dynamics of soil water in podzolic soils in the Scottish Highlands that are then related to surface vegetation and aspect. The authors sampled soils monthly and analzyed the δ18O and δD stable isotope values of soil water from four different depths for one year using an isotopic equilibrium method. The authors inferred evaporative losses from the soil water profile as well as explored relationships between isotope fractionation and deuterium excess (normalized to the local meteoric water line) with soil organic matter characteristics and corresponding vegetation (forest and heather). They hypothesized that their isotopic patterns were related to precipitation inputs and mixing processes in the soil. They found the unique characteristics of the research site to dominate the isotopic patterns they detected, especially at the near surface. In particular, the high organic matter content of the soil served as an important storage pool that potentially dampened an evaporative signal in the water isotopic patterns. During summer, when evaporative potential is high, the evaporative signal was strongest in the upper 15 cm.

The experimental design is tailored for testing vegetation influence on site hydrology, the methods are appropriate and the analyses were exhaustive. However, a justification or more information is needed regarding the use of hydrocalculator. The issue at hand is that at the core of hydrocalculator is the Craig-Gordon model, which is designed for open water surfaces – which the soil clearly is not. There are other models that consider diffusion in the unsaturated zone (i.e., Barnes and Allison, Journal of Hydrology, 1988) and they are able to model the profile while estimating an evaporative flux. In the study under consideration, estimating evaporation is not the focus per se, so removing these potentially erroneous estimates wouldn't be a problem. The figures are in general useful, however, I am afraid figure 8 is too busy to extract the point being addressed without investing a significant amount of time.

*Response: We agree that the application of the Hydrocalculator is a first approximation and uncertain, since it was developed for isotope mass balances of open waters. We therefore revisited the evaporation estimates and changed the approach as follows to justify its application.*

*We do not use the Hydrocalculator anymore, where the exponent of the diffusion coefficient ratio (see Eq. 1 in Horita et al. (2008) or Eq. 6b in Gat (1996)) is assumed to be n = 0.5, but use n = 1, which is more representative for more stagnant interfaces like soil water (Horita et al., 2008).*

*We further now calculate the evaporation losses as a fraction of the original water source, rather than correcting the soil water isotope signal for seasonal variable precipitation input (as criticized in a comment by Referee #1 further below). We calculated the isotopic composition of the original water source as the intercept between the evaporation line of the soil water isotope data in the dual-isotope space and the local meteoric water line according to Javaux et al. (2016). This isotopic signal represents $\delta_P$ in Eq. 3 of our manuscript and the measured soil water isotopes represent $\delta_S$. We are grateful for the suggestion to weight the isotopic composition according to the soil moisture and applied this in the revised calculations.*

*In conclusion, we believe that this revised approach to estimate the fraction of evaporated water is a better representation of the physical processes and we thank Referee #1 for their suggestions.*

*We are aware of the methods discussed by Barnes and Allison (1988), but we think that their presented approaches cannot be applied to our setting. The wet environment in the Scottish Highlands prevents a development of exponentially shaped $\delta^2 H$ depth profiles. Quoting Barnes and Allison (1988): "infiltration of rainfall in this period will invalidate the approach (page 169)".*

*As discussed in section 4.3 and in accordance to Braud et al. (2005), we believe that a representation of soil water isotope concentration profiles needs to account for non-steady conditions and time variant atmospheric boundary conditions. However, we will add the following sentence in section 4.3: "Potential transient numerical modelling approaches that account for isotopic fractionation of the soil water isotopes are available (Braud et al., 2005; Rothfuss et al., 2012; Mueller et al., 2014)."*

*We also decided to take out Figure 8, since the graph is relatively difficult to understand and Referee #2 asked for shortening the manuscript.*

Despite the issue of the evaporation estimates, it is clear that evaporation occurs within these soils and the role of vegetation is highly relevant. And while this is interesting, it is not entirely unexpected. Emphasis is also placed on the high frequency of the sampling (11 campaigns), but with advent of portable CRDS lasers even monthly samples are considered a rather low frequency sampling strategy (for example, see Volkmann et al., New Phytologist, 2016). Modelling is mentioned in the discussion (L198-213; pg- 22-23) although no modelling is performed and the results from the study are not really brought into the modelling discussion directly. There is a long history of investigating and modelling the soil water isotopic profile, and again while the results are interesting for this particular site, the title seems to promise more than the study can deliver. The merit in the study lies in the site-specific nuances such as the role of organic matter in effecting soil water capacity and the subsequent isotopic mixing that occurs.

*Response: To our knowledge, there has not been an investigation of isotopic fractionation dynamics in pore waters over an entire year covered by almost monthly (11 sampling campaigns) sampled at the same date at four locations in parallel. In addition, our study site is a headwater catchment, with relatively difficult access, as well as being representative of other low energy, humid northern environments. We therefore believe that the presented results are not of limited interest restricted only to this particular study site, but are more widely relevant for stable isotope dynamics in soils of humid northern environments that have not yet been well studied.*

*Despite the new possibilities that come with in-situ measurements of pore water stable isotopes, only the recently published study by Oerter and Bowen (2017) applied it to cover almost one year limited to one particular location. However, for the most part, experimental studies used the high frequency sampling with in-situ stable isotope analysis are short term investigations covering few days (Volkmann et al., 2016; Beyer et al., 2016). Our motivation of the experimental set up was to cover the dynamics that occur within an entire year at a number of dominant landscape units. We will include the discussion above in the introduction of the revised manuscript. We will also include in the introduction the following to show the importance of studying the seasonal variability of soil water isotopes: "While studies based on two sampling campaigns (Goldsmith et al., 2012; Evaristo et al., 2016) were supportive of the ecohydrological separation, as presented by Brooks et al. (2010), newly published work with higher temporal resolution of*

*soil water and xylem water isotope sampling suggests that there are seasonal differences with regard to ecohydrological separation (McCutcheon et al., 2016; Hervé-Fernández et al., 2016)."*

Specific comments: Title: It is not exactly clear what is meant by "the soil-plant- atmosphere interface of the critical zone". In this study soil water is measured, why not just state this?

*Response: We consider changing the title to: Soil water stable isotope reveal evaporation dynamics at the soil-plant-atmosphere interface of the critical zone.*

Abstract: Page1, L13: Because this paper is not a test of the method, it is necessary to report it here.

*Response: We would prefer to keep this information in here, since recent findings (see Orlowski et al. (2016)) showed that different methods can potentially result in different findings.*

Page1, L22-25: I would argue that this sentence can be deleted.

*Response: We would not agree with that as we aim to provide a wider relevance of the findings in that sentence.*

Introduction:

Page1, L30: remove "well" from well understood so that it reads simply "insufficiently understood"

*Response: Will be changed as suggested.*

P2 L2: I think it is the age distribution of the water that is used in evapotranspiration that is meant here, and not the age of the flux.

*Response: Yes, changed to "evaporating water".*

P2 L10: perhaps introduce the term isotopic fractionation here, not all readers will understand the relevance of this process.

*Response: We were hoping that the lines 10 to 15 would introduce the term of isotopic fractionation. We will rephrase the sentence to: "However, evaporation leads to isotopic fractionation, where…"*

P2 L32: This sentence needs to be restructured.

*Response: We will change the sentence as follows: However, recent findings about kinetic fractionation in the water pools and tracks of an extended drainage network in a raised bog within the Bruntland Burn showed that evaporation can have a fractionating effect on the stable isotopes of peatland waters, despite the relatively low energy available (Sprenger et al. 2017) ."*

Page 3: Research question 1: Instead of "critical zone", I suggest soil profile.

*Response: Will be changed as suggested.*

Research question 2: This has already been done before. Better to restate your question/goal as to estimate evaporation of the site based on the water stable isotope values within the soil profile. This section might need to be removed if the hydrocalculator approach cannot be justified.

*Response: We are not aware of a similar study that observes the evaporation fractionation in the field at this temporal resolution over a year for four sites. Therefore, we see a clear research gap in understanding the dynamics of soil evaporation. We hope that we better highlight this research gap with the following statement in the introduction: "The temporal variability of the isotopic fractionation in the field has not yet been studied. While Rothfuss et al. (2015) sampled the soil water isotopes in a soil column undergoing evaporation in the laboratory, field studies are usually limited to few sampling campaigns or short period (Twining et al., 2006; Gaj et al., 2016). Additionally, soil...". We will slightly change the research question as follows: "How can one infer soil evaporation dynamics from measured soil water isotopic fractionation?" As discussed above, we believe that the revised estimates of the fraction of evaporation losses are a better representation of the physical processes and should therefore remain as part of the manuscript.*

Research question 3: This is very vague. What is meant by feedbacks? And, how do you expect them to vary spatially and temporally?

*Response: We will change the research question to be more specific as follows: "How do soil characteristics, vegetation cover and aspect drive evaporation fractionation dynamics?*

P3 L20: catchment "is" covered

*Response: Will be changed as suggested.*

Page 5, lines 21-25: Please briefly explain how the 'equilibrium method' works.

*Response: We will include more details on the applied method as suggested.*

P5 L26: Please explain how the standard water was "sampled"?

*Response: In the newly included information on the applied method, we will also add the info that the standard waters were sampled the same way as the soil samples. 10ml of standard water was added into an airtight bag and then heat sealed and allowed to equilibrate for 2 days. The analysis was done the same way as for the soil samples.*

P7 L3: Please discuss the "effect of antecedent conditions". What might we expect that occurs during the time window you suggest?

*Response: We will clarify that we decided to investigate the antecedent condition (in terms of precipitation input averages over 7 and 30 days and PET over 30 days) as follows in the revised manuscript: "To understand the potential atmospheric drivers for the soil water isotopic composition, we investigated the effect of antecedent conditions. We calculated average values of PET over 30 days prior to each sampling campaign ($PET_{30}$) to account for the potential soil evaporation dynamics. Additionally, the precipitation sums and the amount weighted isotopic signal of the daily precipitation isotope samples were computed for the 7 days and 30 days period prior to the sampling ($P_7$ and $P_{30}$, respectively) to assess the mixing processes. Weekly and monthly averages were chosen to see if the relatively young water input or the average over the last month better relate to the observed soil water isotopic signal."*

Page 7, Lines 4-6: Could you briefly explain why exactly you used these number of days (7 and 30)? Furthermore, how were the isotopic signals weighted? This whole approach is rather unclear.

*Response: Please see response above.*

Page 7, Line 13: Did you mix the isotopic results from 5 and 10 cm for achieving f?

*Response: Thanks, this is a good point. We have now weighted the 0-5 cm and 5-10 cm samples by their gravimetric water content.*

Page 7, Lines 15-17: I have some concerns about this approach. What is the purpose of applying this correction to δS? This is not fully clear. Secondly, did you correct the δP for the net precipitation (mm), which was different between from month to month? This would affect the average δ18O and δ2H of precipitation.

*Response: Please see response above. This section will be changed according to the above outlined revised approach to estimate the fraction of evaporation losses.*

P8 L16-17: What is the reason for testing for significance along such a small range? Isn't 0 and 1‰ already in the measurement precision limits? What value is relevant to determine significant evaporation?

*Response: We will add in section 2.5 the following to account for the measurement precision of lc-excess: "The accuracy for the liquid water isotope analysis and the soil water isotope analysis result in a precision limit for lc-excess of about 1.1 ‰ and 3.4 ‰, respectively.*

*Given that the Figure 8 is difficult to understand, while the differences between few permill are not that relevant for the evaporation dynamics considering the measurement limits, we will take this graph out. In the text, we will refer to if lc-excess is significant different from +-3.4 ‰.*

P8 L21: Pearson not Parson

*Response: Will be changed as suggested.*

Figure 3. I really like this figure, especially with the temporal patterns along the secondary plots.

*Response: Thanks, this way of presenting additionally boxplots to the dual isotope plot was inspired by Hervé-Fernández et al. (2016).*

Figure 5 b. The pattern in lc-excess over the season is interesting but what qualifies this as hysteresis and not just seasonal changes in PET? This plot needs error bars along both axes.

*Response: We do not see how the variability of PET as shown with error bars along the x-axis is relevant for the interpretation of the data. Error bars representing the standard deviation of PET 30 days prior to sampling would surrogate that the PET is relatively variable. But instead, we chose the 30 day average to represent the long term variability of the PET and not short term variability. Therefore, we do not see how error bars of the standard deviation of PET over 30 days prior to the sampling would improve the Figure. We show here the Figure with error bars representing the standard deviation of lc-excess within the data set for each sampling day. The information gained is relatively small, since the standard deviation does not vary in time. We would therefore prefer to keep the original figure. The high variability within the depth profiles is shown in Figure 9.*

[Figure]

Page 14, Lines 22-23: What is 'organic content'? Maybe "organic matter content" is more appropriate.

*Response: We will change as suggested for the entire manuscript.*

Figure 7. Is there a reason why LOI should change in such a short time period? The x label of this figure is really confusing.

*Response: In Figure 7, we assume that LOI stays relatively constant over time. The correlation between LOI and GWC or lc-excess is only due to changes in GWC or lc-excess. We hope that the x-axis is better understandable as "Antecedent dry days [%]" and the following figure caption: "Relationship between coefficient of determination of the relationship between lc-excess and LOI (grey) and GWC and LOI (blue) with the percentage of dry days during the 30 day period prior to the soil sampling."*

[Figure]

P16 L74: I think this is in reference to 0‰ lc-excess

*Response: That is correct and we will include the missing information as follows: "The soil water lc-excess at 15 – 20 cm was usually not…"*

Page 17 L85: Should it be "depths" instead of "sites"?

*Response: Thanks, this should be depths and will be changed accordingly.*

Discussion:

P21 L159: Be careful, technically even precipitation has undergone fractionation!

*Response: We will rephrase this sentence for clarification as follows: "With increasing new precipitation input (lc-excess close to zero), the sites…"*

P21 L170: I assume the dynamics at 5cm are being described here. Is the replacement of autumn water the only possibility here? Is it not also possible that mixing (which the authors advocate elsewhere) is responsible? Are they mutually exclusive?

*Response: Mixing and replacement is not mutually exclusive, since both will happen in parallel, but it is difficult to assess which process is dominating. We changed the sentence to: "However, the significantly lower $\delta^2H$ values in the top 5 cm indicate that, despite the potential occurrence of preferential flow, most of the depleted precipitation input was stored in the very top soil and the more enriched soil water from autumn was partly mixed with and partly replaced by the event water."*

P21 L171: "further special" is a bit awkward

*Response: We will replace "further" with "also".*

P22 L172-174: Time reference is lost here. Is the rest of the year for any time that does not occur in January? It looks like this pattern was also emerging in November of 2015 (probably also established in December).

*Response: Yes, we refer here to the vegetation period and replace therefore "rest of the year" with "vegetation period".*

P22 L190-197: I don't think this part of the discussion warrants a whole paragraph.

*Response: We believe that this paragraph is highly relevant, given the recent call for higher spatial soil sampling to address the Two Water World Hypothesis (Berry et al., 2017). We therefore rephrased the paragraph as follows and hope that the Referee agrees that this paragraph is relevant: "In contrast to Geris et al. (2015), who sampled the soil-vegetation units with duplicates, our sampling design with five replicates allowed for a clearer assessment of the spatial heterogeneity of the subsurface. For example, the standard deviation of the SW $\delta^2H$ and $\delta^{18}O$ values for the 5 cm depth increments at each site was - for the entirely sampled upper 20 cm of soil - always higher than the measurement accuracy of 1.13 ‰ and 0.31 ‰, respectively. At Bruntland Burn – as in most Northern temperate and boreal biomes (Jackson et al., 1996)- topsoils contain almost all the root biomass. For the interpretation of potential sources of root water uptake this means that the uncertainty of the potential water source signal due to the heterogenous isotopic composition at particular depths within the rooting zone is higher than the error due to the measurements. Our field measurements underline, therefore, the need for an improved spatial resolution of soil water sampling when studying root water uptake patterns with stable isotopes, as recently called*

*for by Berry et al. (2017). Further, this high variability will potentially impact the application of soil water isotopes for the calibration of soil physical models and the resulting interpretation (Sprenger et al., 2015b). "*

Page 22, Lines 192-195: This statement is not fully clear. What is the importance of the accuracy? Please remember that you are only talking about the top 5 cm. So root water uptake will depend on the plant species and rooting depth.

*Response: Please see above comment and changes of the paragraph.*

P23 L217-218: "kinetic fractionation dynamics" is a little cumbersome; it may be easier to the reader if you simply refer to evaporation when referencing the isotopic effect.

*Response: We will replace "kinetic fractionation" by "soil evaporation".*

P24 L246: citation

*Response: We combined the two sentences so that (Sprenger et al., 2015a) is the reference for the mobility of water sampled with the two different methods.*

P24 L255: Do you mean isotopically depleted infiltration water? I don't immediately see how this study shows that the "the legacy of evaporation losses" allows for separating pools of different water mobility in the SPA interface. Can you make this more apparent?

*Response: We added: "This means that old (more tightly bound) water might not only have a distinct $\delta^2H$ or $\delta^{18}O$ signal compared to mobile water due to seasonally variable precipitation inputs, but also evaporative enrichment signal from periods of high soil evaporation. In conclusion, when relating isotope values of xylem water to soil water to study root water uptake patterns, an evaporation signal (that is lc-excess of xylem water < 0) would not be paradoxical, but simply represent the range of available soil water in the subsurface."*

P25 L279: Where is the number 3 coming from? I think a few citations or reviews might help back this up. I think it is also important to keep the context of the research question in mind when assessing another study's design.

*Response: We agree that the research question should be considered. We are referring here to studies that investigate the root water uptake patterns and do not cover the temporal variability of soil water isotope dynamics. We will also refer now to the Berry et al. (2017) who called for a higher temporal resolution of soil water isotope sampling when investigating root water uptake pattern with stable isotopes. "In line with the call for a higher temporal resolution of soil water isotope sampling (Berry et al., 2017), the highly dynamic isotopic signal during the transition between the dormant and growing seasons underlines the importance of not limiting the soil water isotope sampling to a few sampling campaigns (usually n ≤ 3 in Brooks et al. (2010), Evaristo et al. (2016), Goldsmith et al. (2012)), when investigating root water uptake patterns."*

P25 L288: Isn't the storage capacity referred to earlier relevant here as well?

*Response: Yes, we meant to refer to the storage capacity differences here and clarify this by changing the sentence to: "However, these differences in water storage capacity seemed to only influence the*

*evaporation signal during periods of high precipitation input, when evaporation is already likely to be low (Figure 7)."*

P25 L302-303: citation

*Response: We will add (Allison et al., 1983; Barnes and Allison, 1988) as follows: "The fractionation signal was shown to be more pronounced for evaporation losses from drier soils compared to wetter soils (Allison et al., 1983; Barnes and Allison, 1988)."*

P26 L345: I don't think the word "exceptionally" is warranted here, although the study is data rich. This sentence is also structured in a strange manner (i.e., "but also" when there isn't a contrast to begin with). I don't think the final statement is justified. Are modelers calling for higher spatial and temporal isotope data? Is this listed as a research priority in the literature? Do the data presented here help realize a realistic representation of "soil-vegetation" interactions? Statements like these are beyond the scope of this study.

*Response: We will split this into two sentences and will rephrase them for more clarity. We further will include Vereecken (2016) and McDonnell and Beven (2014) as reference, who are calling for soil water isotope data in high temporal and spatial resolution to calibrate soil physical models and isotope tracer data from within the catchments to foster process understanding, respectively. "The presented soil water isotope data covering the seasonal dynamics at high spatial resolution (5 cm increments at four locations) will allow us to test efficiencies of soil physical models in simulating the water flow and transport in the critical zone (as called for by Vereecken (2016)). The data can further provide a basis to benchmark hydrological models for a more realistic representation of the celerities and velocities when simulating water fluxes and their ages within catchments (McDonnell and Beven (2014)).*

**References**

Allison, G., Barnes, C., and Hughes, M.: The distribution of deuterium and $^{18}$O in dry soils 2. Experimental, Journal of Hydrology, 64, 377–397, doi:10.1016/0022-1694(83)90078-1, 1983.

Barnes, C. and Allison, G.: Tracing of water movement in the unsaturated zone using stable isotopes of hydrogen and oxygen, Journal of Hydrology, 100, 143–176, doi:10.1016/0022-1694(88)90184-9, 1988.

Berry, Z. C., Evaristo, J., Moore, G., Poca, M., Steppe, K., Verrot, L., Asbjornsen, H., Borma, L. S., Bretfeld, M., Hervé-Fernández, P., Seyfried, M., Schwendenmann, L., Sinacore, K., Wispelaere, L. de, and McDonnell, J.: The two water worlds hypothesis: Addressing multiple working hypotheses and proposing a way forward, Ecohydrol., doi:10.1002/eco.1843, 2017.

Beyer, M., Koeniger, P., Gaj, M., Hamutoko, J. T., Wanke, H., and Himmelsbach, T.: A deuterium-based labeling technique for the investigation of rooting depths, water uptake dynamics and unsaturated zone water transport in semiarid environments, J Hydrol, 533, 627–643, doi:10.1016/j.jhydrol.2015.12.037, 2016.

Braud, I., Bariac, T., Gaudet, J. P., and Vauclin, M.: SiSPAT-Isotope, a coupled heat, water and stable isotope (HDO and H2$^{18}$O) transport model for bare soil. Part I. Model description and first verifications, Journal of Hydrology, 309, 277–300, doi:10.1016/j.jhydrol.2004.12.013, 2005.

Brooks, J. R., Barnard, H. R., Coulombe, R., and McDonnell, J. J.: Ecohydrologic separation of water between trees and streams in a Mediterranean climate, Nat Geosci, 3, 100–104, doi:10.1038/NGEO722, 2010.

Evaristo, J., McDonnell, J. J., Scholl, M. A., Bruijnzeel, L. A., and Chun, K. P.: Insights into plant water uptake from xylem-water isotope measurements in two tropical catchments with contrasting moisture conditions, Hydrol. Process., 30, 3210–3227, doi:10.1002/hyp.10841, 2016.

Gaj, M., Beyer, M., Koeniger, P., Wanke, H., Hamutoko, J., and Himmelsbach, T.: In situ unsaturated zone water stable isotope ($^2$H and $^{18}$O) measurements in semi-arid environments: A soil water balance, Hydrol. Earth Syst. Sci., 20, 715–731, doi:10.5194/hess-20-715-2016, 2016.

Gat, J. R.: Oxygen and hydrogen isotopes in the hydrologic cycle, Annu. Rev. Earth Planet. Sci., 24, 225–262, doi:10.1146/annurev.earth.24.1.225, 1996.

Geris, J., Tetzlaff, D., McDonnell, J., and Soulsby, C.: The relative role of soil type and tree cover on water storage and transmission in northern headwater catchments, Hydrol. Process., 29, 1844–1860, doi:10.1002/hyp.10289, 2015.

Goldsmith, G. R., Muñoz-Villers, L. E., Holwerda, F., McDonnell, J. J., Asbjornsen, H., and Dawson, T. E.: Stable isotopes reveal linkages among ecohydrological processes in a seasonally dry tropical montane cloud forest, Ecohydrol., 5, 779–790, doi:10.1002/eco.268, 2012.

Hervé-Fernández, P., Oyarzún, C., Brumbt, C., Huygens, D., Bodé, S., Verhoest, N. E. C., and Boeckx, P.: Assessing the "two water worlds" hypothesis and water sources for native and exotic evergreen species in south-central Chile, Hydrol. Process., 30, 4227–4241, doi:10.1002/hyp.10984, 2016.

Horita, J., Rozanski, K., and Cohen, S.: Isotope effects in the evaporation of water: a status report of the Craig-Gordon model, Isotopes in Environmental and Health Studies, 44, 23–49, doi:10.1080/10256010801887174, 2008.

Jackson, R. B., Canadell, J., Ehleringer, J. R., Mooney, H. A., Sala, O. E., and Schulze, E. D.: A global analysis of root distributions for terrestrial biomes, Oecologia, 108, 389–411, doi:10.1007/BF00333714, 1996.

Javaux, M., Rothfuss, Y., Vanderborght, J., Vereecken, H., and Bruggemann, N.: Isotopic composition of plant water sources, Nature, 536, E1-3, doi:10.1038/nature18946, 2016.

McCutcheon, R. J., McNamara, J. P., Kohn, M. J., and Evans, S. L.: An Evaluation of the Ecohydrological Separation Hypothesis in a Semiarid Catchment, Hydrol. Process., 31, 783–799, doi:10.1002/hyp.11052, 2016.

McDonnell, J. J. and Beven, K.: Debates-The future of hydrological sciences: A (common) path forward? A call to action aimed at understanding velocities, celerities and residence time distributions of the headwater hydrograph, Water Resour. Res., 50, 5342–5350, doi:10.1002/2013WR015141, 2014.

Mueller, M. H., Alaoui, A., Kuells, C., Leistert, H., Meusburger, K., Stumpp, C., Weiler, M., and Alewell, C.: Tracking water pathways in steep hillslopes by $\delta^{18}$O depth profiles of soil water, J Hydrol, 519, 340–352, doi:10.1016/j.jhydrol.2014.07.031, 2014.

Oerter, E. and Bowen, G.: In situ monitoring of H and O stable isotopes in soil water reveals ecohydrologic dynamics in managed soil systems, Ecohydrol., doi:10.1002/eco.1841, 2017.

Orlowski, N., Pratt, D. L., and McDonnell, J. J.: Intercomparison of soil pore water extraction methods for stable isotope analysis, Hydrol. Process., 30, 3434–3449, doi:10.1002/hyp.10870, 2016.

Rothfuss, Y., Braud, I., Le Moine, N., Biron, P., Durand, J.-L., Vauclin, M., and Bariac, T.: Factors controlling the isotopic partitioning between soil evaporation and plant transpiration: Assessment

using a multi-objective calibration of SiSPAT-Isotope under controlled conditions, Journal of Hydrology, 442–443, 75–88, doi:10.1016/j.jhydrol.2012.03.041, 2012.

Rothfuss, Y., Merz, S., Vanderborght, J., Hermes, N., Weuthen, A., Pohlmeier, A., Vereecken, H., and Brüggemann, N.: Long-term and high-frequency non-destructive monitoring of water stable isotope profiles in an evaporating soil column, Hydrol. Earth Syst. Sci., 19, 4067–4080, doi:10.5194/hess-19-4067-2015, 2015.

Sprenger, M., Herbstritt, B., and Weiler, M.: Established methods and new opportunities for pore water stable isotope analysis, Hydrol. Process., 29, 5174–5192, doi:10.1002/hyp.10643, 2015a.

Sprenger, M., Volkmann, T. H. M., Blume, T., and Weiler, M.: Estimating flow and transport parameters in the unsaturated zone with pore water stable isotopes, Hydrol. Earth Syst. Sci., 19, 2617–2635, doi:10.5194/hess-19-2617-2015, 2015b.

Twining, J., Stone, D., Tadros, C., Henderson-Sellers, A., and Williams, A.: Moisture Isotopes in the Biosphere and Atmosphere (MIBA) in Australia: A priori estimates and preliminary observations of stable water isotopes in soil, plant and vapour for the Tumbarumba Field Campaign, Global and Planetary Change, 51, 59–72, doi:10.1016/j.gloplacha.2005.12.005, 2006.

Vereecken, H.: Modeling Soil Processes: Review, Key challenges and New Perspectives, Vadose Zone J, 15, 1–57, 2016.

Volkmann, T. H. M., Haberer, K., Gessler, A., and Weiler, M.: High-resolution isotope measurements resolve rapid ecohydrological dynamics at the soil-plant interface, New Phytol, 210, 839–849, doi:10.1111/nph.13868, 2016.

---

## Author Response (AR1)

**Editor's comments to the Author:**

Dear Dr Sprenger and co-authors,

Based on two reviews of your paper, I recommend that it may be suitable for publication after major revisions. Please consider the reviewers comments carefully in your revision. In particular, it is recommended that the paper is significantly shortened, including a reduction in the number of figures, and the title is changed to be more accurate.

Kind Regards,

Hilary McMillan

**Response to the Editor**

*Dear Hilary McMillan,*

*Thank you for handling our manuscript and considering the study for publication in HESS after major revision. We addressed all the referee's comments by a reply to each comment and changed the original manuscript accordingly. Please see the replies to the referee's comments below and the track changed manuscript for the major revisions undertaken in the manuscript.*

*As recommended by both referees, we changed the manuscript title to: "Soil water stable isotopes reveal evaporation dynamics at the soil-plant-atmosphere interface of the critical zone", which clarifies that soil water isotopes have been measured, from which we infer evaporation dynamics at the soil-plant-atmosphere interface. We believe that the term "soil-plant-atmosphere interface" is appropriate in the context of our study, since we cover with our sampling design the major rooting depth and thus the part of the critical zone, where the interlinkages between the soil, vegetation and atmosphere takes place.*

*As recommended by Referee #1, we revisited the evaporation estimates using the Craig-Gordon model. We now apply the Craig-Gordon approach adjusted for evaporation from soils using n=1 for the exponent of the diffusion coefficient ratio (see Eq. 1 in Horita et al. (2008) or Eq. 6b in Gat (1996)), which is more representative for more stagnant interfaces like soil water (Horita et al., 2008). We further now calculate the evaporation losses as a fraction of the original water source, rather than correcting the soil water isotope signal for seasonal variable precipitation input (as criticized in a comment by Referee #1). We calculated the isotopic composition of the original water source as the intercept between the evaporation line of the soil water isotope data in the dual-isotope space and the local meteoric water line according to Javaux et al. (2016). This isotopic signal represents $\delta_P$ in Eq. 3 of our manuscript and the measured soil water isotopes represent $\delta_S$. We also included the suggestion of Referee #1 to weight the isotopic composition according to the soil moisture and applied this in the revised calculations.*

*Another major change is the shortening of the manuscript, where we reduced the number of figures by 2 and revised the text with special focus on repetitive paragraphs to shorten the volume.*

*We hope that the revised version meets the Editor's and Referee's concerns and that the study will therefore still be considered for publication in HESS.*

**Response to comments by Anonymous Referee #1**

*We thank Referee #1 for reviewing our paper and their positive feedback on the methods and analysis that we present in our manuscript. We are pleased to see that we could convey the message of the presented study in a convincing way, given the synopsis Referee #1 provides at the beginning of their review. We will first comment on the major issues raised by Referee #1 and further respond to each specific comment by Referee #1 below.*

The manuscript describes evaporation dynamics of soil water in podzolic soils in the Scottish Highlands that are then related to surface vegetation and aspect. The authors sampled soils monthly and analzyed the δ18O and δD stable isotope values of soil water from four different depths for one year using an isotopic equilibrium method. The authors inferred evaporative losses from the soil water profile as well as explored relationships between isotope fractionation and deuterium excess (normalized to the local meteoric water line) with soil organic matter characteristics and corresponding vegetation (forest and heather). They hypothesized that their isotopic patterns were related to precipitation inputs and mixing processes in the soil. They found the unique characteristics of the research site to dominate the isotopic patterns they detected, especially at the near surface. In particular, the high organic matter content of the soil served as an important storage pool that potentially dampened an evaporative signal in the water isotopic patterns. During summer, when evaporative potential is high, the evaporative signal was strongest in the upper 15 cm.

The experimental design is tailored for testing vegetation influence on site hydrology, the methods are appropriate and the analyses were exhaustive. However, a justification or more information is needed regarding the use of hydrocalculator. The issue at hand is that at the core of hydrocalculator is the Craig-Gordon model, which is designed for open water surfaces – which the soil clearly is not. There are other models that consider diffusion in the unsaturated zone (i.e., Barnes and Allison, Journal of Hydrology, 1988) and they are able to model the profile while estimating an evaporative flux. In the study under consideration, estimating evaporation is not the focus per se, so removing these potentially erroneous estimates wouldn't be a problem. The figures are in general useful, however, I am afraid figure 8 is too busy to extract the point being addressed without investing a significant amount of time.

*Response: We agree that the application of the Hydrocalculator is a first approximation and uncertain, since it was developed for isotope mass balances of open waters. We therefore revisited the evaporation estimates and changed the approach as follows to justify its application.*

*We do not use the Hydrocalculator anymore, where the exponent of the diffusion coefficient ratio (see Eq. 1 in Horita et al. (2008) or Eq. 6b in Gat (1996)) is assumed to be $n = 0.5$, but use $n = 1$, which is more representative for more stagnant interfaces like soil water (Horita et al., 2008).*

*We further now calculate the evaporation losses as a fraction of the original water source, rather than correcting the soil water isotope signal for seasonal variable precipitation input (as criticized in a comment by Referee #1 further below). We calculated the isotopic composition of the original water source as the intercept between the evaporation line of the soil water isotope data in the dual-isotope space and the local meteoric water line according to Javaux et al. (2016). This isotopic signal represents $\delta_P$ in Eq. 3 of our manuscript and the measured soil water isotopes represent $\delta_S$. We are grateful for the suggestion to weight the isotopic composition according to the soil moisture and applied this in the revised calculations.*

*In conclusion, we believe that this revised approach to estimate the fraction of evaporated water is a better representation of the physical processes and we thank Referee #1 for their suggestions.*

*We are aware of the methods discussed by Barnes and Allison (1988), but we think that their presented approaches cannot be applied to our setting. The wet environment in the Scottish Highlands prevents a development of exponentially shaped $\delta^2H$ depth profiles. Quoting Barnes and Allison (1988): "infiltration of rainfall in this period will invalidate the approach (page 169)".*

*As discussed in section 4.3 and in accordance to Braud et al. (2005), we believe that a representation of soil water isotope concentration profiles needs to account for non-steady conditions and time variant atmospheric boundary conditions. However, we added the following sentence in section 4.3: "Potential transient numerical modelling approaches that account for isotopic fractionation of the soil water isotopes are available (Braud et al., 2005; Rothfuss et al., 2012; Mueller et al., 2014)."*

*We also decided to take out Figure 8, since the graph is relatively difficult to understand and Referee #2 asked for shortening the manuscript. Furthermore, we did not include Figure 4 in the revised version to shorten the manuscript.*

Despite the issue of the evaporation estimates, it is clear that evaporation occurs within these soils and the role of vegetation is highly relevant. And while this is interesting, it is not entirely unexpected. Emphasis is also placed on the high frequency of the sampling (11 campaigns), but with advent of portable CRDS lasers even monthly samples are considered a rather low frequency sampling strategy (for example, see Volkmann et al., New Phytologist, 2016). Modelling is mentioned in the discussion (L198-213; pg- 22-23) although no modelling is performed and the results from the study are not really brought into the modelling discussion directly. There is a long history of investigating and modelling the soil water isotopic profile, and again while the results are interesting for this particular site, the title seems to promise more than the study can deliver. The merit in the study lies in the site-specific nuances such as the role of organic matter in effecting soil water capacity and the subsequent isotopic mixing that occurs.

*Response: To our knowledge, there has not been an investigation of isotopic fractionation dynamics in pore waters over an entire year covered by almost monthly (11 sampling campaigns) sampled at the same date at four locations in parallel. In addition, our study site is a headwater catchment, with relatively difficult access, as well as being representative of other low energy, humid northern environments. We therefore believe that the presented results are not of limited interest restricted only to this particular study site, but are more widely relevant for stable isotope dynamics in soils of humid northern environments that have not yet been well studied.*

*Despite the new possibilities that come with in-situ measurements of pore water stable isotopes, only the recently published study by Oerter and Bowen (2017) applied it to cover almost one year limited to one particular location. However, for the most part, experimental studies used the high frequency sampling with in-situ stable isotope analysis are short term investigations covering few days (Volkmann et al., 2016; Beyer et al., 2016). Our motivation of the experimental set up was to cover the dynamics that occur within an entire year at a number of dominant landscape units. We will include the discussion above in the introduction of the revised manuscript. We will also include in the introduction the following to show the importance of studying the seasonal variability of soil water isotopes: "While studies based on two sampling campaigns (Goldsmith et al., 2012; Evaristo et al., 2016) were supportive of the ecohydrological separation, as presented by Brooks et al. (2010), newly published work with higher temporal resolution of*

*soil water and xylem water isotope sampling suggests that there are seasonal differences with regard to ecohydrological separation (McCutcheon et al., 2016; Hervé-Fernández et al., 2016)."*

Specific comments: Title: It is not exactly clear what is meant by "the soil-plant- atmosphere interface of the critical zone". In this study soil water is measured, why not just state this?

*Response: We changed the title to: Soil water stable isotope reveal evaporation dynamics at the soil-plant-atmosphere interface of the critical zone.*

Abstract: Page1, L13: Because this paper is not a test of the method, it is necessary to report it here.

*Response: We would prefer to keep this information in here, since recent findings (see Orlowski et al. (2016)) showed that different methods can potentially result in different findings.*

Page1, L22-25: I would argue that this sentence can be deleted.

*Response: We would not agree with that as we aim to provide a wider relevance of the findings in that sentence.*

Introduction:

Page1, L30: remove "well" from well understood so that it reads simply "insufficiently understood"

*Response: Changed as suggested.*

P2 L2: I think it is the age distribution of the water that is used in evapotranspiration that is meant here, and not the age of the flux.

*Response: Yes, changed to "evaporating water".*

P2 L10: perhaps introduce the term isotopic fractionation here, not all readers will understand the relevance of this process.

*Response: We were hoping that the lines 10 to 15 would introduce the term of isotopic fractionation. We rephrased the sentence to: "However, evaporation leads to isotopic fractionation, where…"*

P2 L32: This sentence needs to be restructured.

*Response: We changed the sentence as follows: "However, recent findings about kinetic fractionation in the water pools and tracks of an extended drainage network in a raised bog within the Bruntland Burn showed that evaporation can have a fractionating effect on the stable isotopes of peatland waters, despite the relatively low energy available (Sprenger et al. 2017)."*

Page 3: Research question 1: Instead of "critical zone", I suggest soil profile.

*Response: Changed as suggested.*

Research question 2: This has already been done before. Better to restate your question/goal as to estimate evaporation of the site based on the water stable isotope values within the soil profile. This section might need to be removed if the hydrocalculator approach cannot be justified.

*Response: We are not aware of a similar study that observes the evaporation fractionation in the field at this temporal resolution over a year for four sites. Therefore, we see a clear research gap in understanding the dynamics of soil evaporation. We hope that we better highlight this research gap with the following statement in the introduction: "The temporal variability of the isotopic fractionation in the field has not yet been studied. While Rothfuss et al. (2015) sampled the soil water isotopes in a soil column undergoing evaporation in the laboratory, field studies are usually limited to few sampling campaigns or short period (Twining et al., 2006; Gaj et al., 2016). Additionally, soil…". We changed the research question as follows: "How can one infer soil evaporation dynamics from measured soil water isotopic fractionation?" As discussed above, we believe that the revised estimates of the fraction of evaporation losses are a better representation of the physical processes and should therefore remain as part of the manuscript.*

Research question 3: This is very vague. What is meant by feedbacks? And, how do you expect them to vary spatially and temporally?

*Response: We changed the research question to be more specific as follows: "How do soil characteristics, vegetation cover and aspect drive evaporation fractionation dynamics?*

P3 L20: catchment "is" covered

*Response: Changed as suggested.*

Page 5, lines 21-25: Please briefly explain how the 'equilibrium method' works.

*Response: We included more details on the applied method as suggested.*

P5 L26: Please explain how the standard water was "sampled"?

*Response: In the newly included information on the applied method, we also added the info that the standard waters were sampled the same way as the soil samples. 10 ml of standard water was added into an airtight bag and then heat sealed and allowed to equilibrate for 2 days. The analysis was done the same way as for the soil samples.*

P7 L3: Please discuss the "effect of antecedent conditions". What might we expect that occurs during the time window you suggest?

*Response: We clarified that we decided to investigate the antecedent condition (in terms of precipitation input averages over 7 and 30 days and PET over 30 days) as follows in the revised manuscript: "To understand the potential atmospheric drivers for the soil water isotopic composition, we investigated the effect of antecedent conditions. We calculated average values of PET over 30 days prior to each sampling campaign ($PET_{30}$) to account for the potential soil evaporation dynamics. Additionally, the precipitation sums and the amount weighted isotopic signal of the daily precipitation isotope samples were computed for the 7 days and 30 days period prior to the sampling ($P_7$ and $P_{30}$, respectively) to assess the mixing processes. Weekly and monthly averages were chosen to see if the relatively young water input or the average over the last month better relate to the observed soil water isotopic signal."*

Page 7, Lines 4-6: Could you briefly explain why exactly you used these number of days (7 and 30)? Furthermore, how were the isotopic signals weighted? This whole approach is rather unclear.

*Response: Please see response above.*

Page 7, Line 13: Did you mix the isotopic results from 5 and 10 cm for achieving f?

*Response: Thanks, this is a good point. We have now weighted the 0-5 cm and 5-10 cm samples by their gravimetric water content.*

Page 7, Lines 15-17: I have some concerns about this approach. What is the purpose of applying this correction to δS? This is not fully clear. Secondly, did you correct the δP for the net precipitation (mm), which was different between from month to month? This would affect the average δ18O and δ2H of precipitation.

*Response: Please see response above. This section was changed according to the above outlined revised approach to estimate the fraction of evaporation losses.*

P8 L16-17: What is the reason for testing for significance along such a small range? Isn't 0 and 1‰ already in the measurement precision limits? What value is relevant to determine significant evaporation?

*Response: We added in section 2.5 the following to account for the measurement precision of lc-excess: "The accuracy for the liquid water isotope analysis and the soil water isotope analysis result in a precision limit for lc-excess of about 1.1 ‰ and 3.4 ‰, respectively.*

P8 L21: Pearson not Parson

*Response: Changed as suggested.*

Figure 3. I really like this figure, especially with the temporal patterns along the secondary plots.

*Response: Thanks, this way of presenting additionally boxplots to the dual isotope plot was inspired by Hervé-Fernández et al. (2016).*

Figure 5 b. The pattern in lc-excess over the season is interesting but what qualifies this as hysteresis and not just seasonal changes in PET? This plot needs error bars along both axes.

*Response: We do not see how the variability of PET as shown with error bars along the x-axis is relevant for the interpretation of the data. Error bars representing the standard deviation of PET 30 days prior to sampling would surrogate that the PET is relatively variable. But instead, we chose the 30 day average to represent the long term variability of the PET and not short term variability. Therefore, we do not see how error bars of the standard deviation of PET over 30 days prior to the sampling would improve the Figure. We show here the Figure with error bars representing the standard deviation of lc-excess within the data set for each sampling day. The information gained is relatively small, since the standard deviation does not vary in time. We would therefore prefer to keep the original figure. The high variability within the depth profiles is shown in Figure 7 of the revised manuscript.*

[Figure]

Page 14, Lines 22-23: What is 'organic content'? Maybe "organic matter content" is more appropriate.

*Response: Change as suggested for the entire manuscript.*

Figure 7. Is there a reason why LOI should change in such a short time period? The x label of this figure is really confusing.

*Response: In Figure 7, we assume that LOI stays relatively constant over time. The correlation between LOI and GWC or lc-excess is only due to changes in GWC or lc-excess. We hope that the x-axis is better understandable as "Antecedent dry days [%]" and the following figure caption: "Relationship between coefficient of determination of the relationship between lc-excess and LOI (grey) and GWC and LOI (blue) with the percentage of dry days during the 30 day period prior to the soil sampling."*

[Figure]

P16 L74: I think this is in reference to 0‰ lc-excess

Response: That is correct and we included the missing information as follows: "The soil water lc-excess at 15 – 20 cm was usually not…"

Page 17 L85: Should it be "depths" instead of "sites"?

Response: Thanks, this should be depths and was changed accordingly.

Discussion:

P21 L159: Be careful, technically even precipitation has undergone fractionation!

Response: We rephraseed this sentence for clarification as follows: *"With increasing new precipitation input (lc-excess close to zero), the sites…"*

P21 L170: I assume the dynamics at 5cm are being described here. Is the replacement of autumn water the only possibility here? Is it not also possible that mixing (which the authors advocate elsewhere) is responsible? Are they mutually exclusive?

*Response: Mixing and replacement is not mutually exclusive, since both will happen in parallel, but it is difficult to assess which process is dominating. We changed the sentence to: "However, the significantly lower $\delta^2H$ values in the top 5 cm indicate that, despite the potential occurrence of preferential flow, most of the depleted precipitation input was stored in the very top soil and the more enriched soil water from autumn was partly mixed with and partly replaced by the event water."*

P21 L171: "further special" is a bit awkward

*Response: We replaced "further" with "also".*

P22 L172-174: Time reference is lost here. Is the rest of the year for any time that does not occur in January? It looks like this pattern was also emerging in November of 2015 (probably also established in December).

*Response: Yes, we refer here to the vegetation period and replaced therefore "rest of the year" with "vegetation period".*

P22 L190-197: I don't think this part of the discussion warrants a whole paragraph.

*Response: We believe that this paragraph is highly relevant, given the recent call for higher spatial soil sampling to address the Two Water World Hypothesis (Berry et al., 2017). We therefore rephrased the paragraph as follows and hope that the Referee agrees that this paragraph is relevant: "In contrast to Geris et al. (2015), who sampled the soil-vegetation units with duplicates, our sampling design with five replicates allowed for a clearer assessment of the spatial heterogeneity of the subsurface. For example, the standard deviation of the SW $\delta^2H$ and $\delta^{18}O$ values for the 5 cm depth increments at each site was - for the entirely sampled upper 20 cm of soil - always higher than the measurement accuracy of 1.13 ‰ and 0.31 ‰, respectively. At Bruntland Burn – as in most Northern temperate and boreal biomes (Jackson et al., 1996)- topsoils contain almost all the root biomass. For the interpretation of potential sources of root water uptake this means that the uncertainty of the potential water source signal due to the heterogenous isotopic composition at particular depths within the rooting zone is higher than the error due to the measurements. Our field measurements underline, therefore, the need for an improved spatial resolution of soil water sampling when studying root water uptake patterns with stable isotopes, as recently called*

*for by Berry et al. (2017). Further, this high variability will potentially impact the application of soil water isotopes for the calibration of soil physical models and the resulting interpretation (Sprenger et al., 2015b).*
*"*

Page 22, Lines 192-195: This statement is not fully clear. What is the importance of the accuracy? Please remember that you are only talking about the top 5 cm. So root water uptake will depend on the plant species and rooting depth.

*Response: Please see above comment and changes of the paragraph.*

P23 L217-218: "kinetic fractionation dynamics" is a little cumbersome; it may be easier to the reader if you simply refer to evaporation when referencing the isotopic effect.

*Response: We replaced "kinetic fractionation" by "soil evaporation".*

P24 L246: citation

*Response: We combined the two sentences so that (Sprenger et al., 2015a) is the reference for the mobility of water sampled with the two different methods.*

P24 L255: Do you mean isotopically depleted infiltration water? I don't immediately see how this study shows that the "the legacy of evaporation losses" allows for separating pools of different water mobility in the SPA interface. Can you make this more apparent?

*Response: We added: "This means that old (more tightly bound) water might not only have a distinct $\delta^2 H$ or $\delta^{18}O$ signal compared to mobile water due to seasonally variable precipitation inputs, but also evaporative enrichment signal from periods of high soil evaporation. In conclusion, when relating isotope values of xylem water to soil water to study root water uptake patterns, an evaporation signal (that is lc-excess of xylem water < 0) would not be paradoxical, but simply represent the range of available soil water in the subsurface."*

P25 L279: Where is the number 3 coming from? I think a few citations or reviews might help back this up. I think it is also important to keep the context of the research question in mind when assessing another study's design.

*Response: We agree that the research question should be considered. We are referring here to studies that investigate the root water uptake patterns and do not cover the temporal variability of soil water isotope dynamics. We also refer now to the Berry et al. (2017) who called for a higher temporal resolution of soil water isotope sampling when investigating root water uptake pattern with stable isotopes. "In line with the call for a higher temporal resolution of soil water isotope sampling (Berry et al., 2017), the highly dynamic isotopic signal during the transition between the dormant and growing seasons underlines the importance of not limiting the soil water isotope sampling to a few sampling campaigns (usually n ≤ 3 in Brooks et al. (2010), Evaristo et al. (2016), Goldsmith et al. (2012)), when investigating root water uptake patterns."*

P25 L288: Isn't the storage capacity referred to earlier relevant here as well?

*Response: Yes, we meant to refer to the storage capacity differences here and clarified this by changing the sentence to: "However, these differences in water storage capacity seemed to only influence the*

*evaporation signal during periods of high precipitation input, when evaporation is already likely to be low (Figure 6)."*

P25 L302-303: citation

*Response: We added references as follows: "The fractionation signal was shown to be more pronounced for evaporation losses from drier soils compared to wetter soils (Allison et al., 1983; Barnes and Allison, 1988)."*

P26 L345: I don't think the word "exceptionally" is warranted here, although the study is data rich. This sentence is also structured in a strange manner (i.e., "but also" when there isn't a contrast to begin with). I don't think the final statement is justified. Are modelers calling for higher spatial and temporal isotope data? Is this listed as a research priority in the literature? Do the data presented here help realize a realistic representation of "soil-vegetation" interactions? Statements like these are beyond the scope of this study.

*Response: We split this into two sentences and will rephrase them for more clarity. We further included Vereecken (2016) and McDonnell and Beven (2014) as reference, who are calling for soil water isotope data in high temporal and spatial resolution to calibrate soil physical models and isotope tracer data from within the catchments to foster process understanding, respectively. "The presented soil water isotope data covering the seasonal dynamics at high spatial resolution (5 cm increments at four locations) will allow us to test efficiencies of soil physical models in simulating the water flow and transport in the critical zone (as called for by Vereecken (2016)). The data can further provide a basis to benchmark hydrological models for a more realistic representation of the celerities and velocities when simulating water fluxes and their ages within catchments (McDonnell and Beven (2014)).*

**Response to comments by Anonymous Referee #2**

*We thank Referee #2 for reviewing our paper, the positive feedback on the visualization of the presented data and that the study will be a good addition to the soil water isotope literature*

*We will first comment on each of the major issues raised by Referee #2 and further respond to each comment by Referee #2 below.*

General comments

The authors studied the influence of vegetation on water fluxes in the upper soil compartment of the Scottish Highlands by means of stable water isotopes. Soil samples were taken eleven times over the course of a year and analyzed for their isotopic composition using the direct equilibration method. The authors nicely visualized their results. However, they should consider cutting down the number of figures.

I think the paper length should also be reduced by at least five pages, which would help focus on the most important points.

*Response: We revised the manuscript with a special focus on repetition of results and discussion to shorten the manuscript. We took out the Figure 4 and Figure 8 of the original manuscript, since also Referee #1 was commenting on the difficulty to understand it. However, Referee #1 asked for a more detailed description of the direct-equilibration method, which will add a few paragraphs to the manuscript.*

There are many repetitions, which unnecessarily blow up the manuscript. What did the authors really expect to find?

*Response: We refer to the literature reviews by Evaristo et al. (2015) and Sprenger et al. (2016) who showed that northern environments have not yet been widely studied regarding their soil water isotope dynamics. Especially the intensity of evaporation fractionation was the main question, since surface waters in an adjacent peatland drainage network showed kinetic a fractionation signal (Sprenger et al. 2017b), but soil water with suction lysimeters in the same experimental catchment showed only limited fractionation (Geris et al. 2015). We therefore believe that the presented results are not just of limited interest to the particular study site, but are more widely and generally relevant for stable isotope dynamics in soils of humid northern environments that have not yet been well studied.*

I think there could also be a more compelling title that illustrates immediately to the potential reader what exactly the paper is about. The title should focus more on the actual findings as the manuscript does not present atmospheric data or detailed data on vegetation (e.g. rooting depth and density) anyway.

*Response: We changed the title to "Soil water stable isotope reveal evaporation dynamics at the soil-plant-atmosphere interface of the critical zone" in order to clarify what compartment was measured. However, we present rooting depth for the heather and present and discuss the atmospheric driver of the isotope dynamics of the soil water.*

The authors pose three research questions. In my opinion, these questions could be more precise. In particular, the third research question cannot really be answered by the results – especially not the atmospheric component.

*Response: We changed the research questions as follows to be more specific:*

*How do precipitation input and the soil water storage mix and affect the soil water isotope dynamics over time?*

*How can one infer soil evaporation dynamics in the field from soil water isotopic fractionation?*

*How do soil characteristics, vegetation cover and aspect drive evaporation fractionation dynamics?*

With regard to the soil samplings, I would not consider the sampling strategy as high frequent, especially against the background of portable laser spectroscopes which can indeed measure water isotopic composition in-situ with high frequency.

*Response: Despite the new possibilities that come with in-situ measurements of pore water stable isotopes, only the recently published study by Oerter, Bowen (2017) applied it to cover almost one year limited to one particular location. However, for the most part, experimental studies used the high frequency sampling with in-situ stable isotope analysis short term investigations covering only a few days (Volkmann*

*et al. 2016; Beyer et al. 2016). Our motivation of the experimental set up was to cover the dynamics that occur within an entire year, capturing seasonal variability at a number of dominant landscape units. We included the discussion above in the introduction of the revised manuscript.*

The authors describe the soil texture of the upper 20 cm as mainly loamy sand. A table, which com- piles all soil properties, would be helpful at this point. Soil properties have been shown to affect the extraction method's isotope results. Do the authors have data on the soil mineralogy (clay mineral composition)? The applied direct equilibration has several downsides: It is less precise for more clayey soils and soils with low water content; storage time is also an issue as it can lead to evaporative water loss through the bag (How long were the bags stored prior to analysis in the present study?) Furthermore, soil organic matter content has been proven to have an effect on gained isotope results. The authors should consider these aspects when discussing their data.

*Response: Please see Table 1 for a detailed list of the soil properties. You see that the clay content was generally very low for the studied soils. As referred to in the manuscript, we have tested our method for soil water isotope analysis for its sensitivity of $CO_2$ emissions during the equilibration period of 2 days. The results have been published elsewhere (Sprenger et al. 2017a). This is why we do not pick up this issue in the discussion. We included in the methods section comments that the conditions at the study site (low clay content and generally high volumetric water content) are in favor of the applied direct-equilibraiton method. We will further mention that the analysis was done within one week after the sampling and that the evaporation losses through the bag can be neglected. A test between different bags (as reported by Sprenger et al. (2015)) showed that the used bag (Weber packaging) loosed less than 0.15% of its stored water over 30 days (details www.hydro.uni-freiburg.de/publ/pubpics/post229).*

In sum, I think this paper will make a good addition to the soil water isotope literature, although it does not contain much novel or surprising findings. However, the authors did a great job in data analyses and presentation.

*Response: Thanks for the positive feedback, however we strongly disagree that the study lacks novelty. We have not found a study that showed the soil water isotope fractionation in the field over an entire year at sites with contrasting land cover; this alone makes our study highly novel.*

Specific comments

P 1 L 13: δ18O

*Response: Changed as suggested.*

P 3 L 1: Thus,. . .

*Response: Changed as suggested.*

P 4 L 14-29: Described in too much detail; consider compiling important soil data in a Table

*Response: We shortened it. Please also see the Table 1.*

P 5 L 3-8: Far too detailed

*Response: We shortened it.*

P 5 L 24: Not necessary to state model and serial number of the isotope analyzer

*Response: Was removed.*

P 6 L 7: Not necessary to reference the python module; please change throughout the manuscript

*Response: Changed as suggested.*

P 8 L 4 –P9 L3: This whole section is again too long. Please condense

*Response: We believe that details of the applied statistical analysis are of relevance for the reader.*

P 9 L 6: Different font used

*Response: Changed as suggested.*

P 11 Fig. 2: for a) I would suggest to plot the rainfall amount data inversely (top-down) and either change the scale of the axis or the size of the blue star so that they are not cut off; for b) consider including moving averages through the soil data (e.g., moving average for the top and subsoil); describe the color code of the soil data (light brown dots stand for. . .)

*Response: We changed the bar plot in a way that the y-axis is inversely and adjust the axis scale to prevent cut off. We will not consider including moving averages, since the physical meaning of such a moving average is questionable, given that we do not know the soil water isotopic composition of up to 30 days between two sampling campaigns.*

P 13 Fig. 4: This figure does not add much information; consider deleting this figure. Does the average precipitation input signal represent a 1-yr mean?

*Response: We removed this figure.*

P 16 chp. 3.2.2.: Include this section in results section 3.1 as it does not add too much new information

*Response: We prefer to keep the structure of the results section with a focus on temporal (3.1) and spatial (3.2) variability. The differences in depth would therefore be part of section 3.2. We removed a paragraphs in 3.1, which dealt with the former Figure 4 and 8. Thus, there is no repetition anymore on the variability over the soil depth.*

P 20 chp. 3.2.4 Delete this section. There are no sig. differences in isotopic signatures when considering the aspect.

*Response: We believe that also a negative result is worth mentioning.*

P 20 L n134 ff: Repetition; consider deleting

*Response: We rephrased this section according to the new evaporation loss estimates.*

P 21 L 167-68: bypass flow, really; not so much differences over depth here

*Response: We agree that it is difficult to infer from hydrometric data (here GWC) to mixing and removed that sentence.*

P 22 L 175 : Is throughfall data available for these sites to underline this statement?

*Response: Yes, we now refer to the throughfall study here as follows: "The higher variability of the SW $\delta^2 H$ values beneath Scots pine compared to the SW beneath heather (Figure 9) cannot be explained by differences of the throughfall isotopic signal, since they are minor for the two vegetation types (Braun 2015). The higher variability in the isotopic signal therefore indicates that flow paths are generally more variable in the forest soils."*

P 24 L 260: Add Gaj et al. (2017b)

*Response: Gaj et al. (2017) was added.*

P 24 L 265: This is not a new finding and not really surprising.

*Response: If this is not surprising, why is it ignored in studies dealing with root water uptake pattern with the means of stable water isotopes?*

P 24 L 269: The authors compare their study with results by Geris et al. (2015a) quite frequently. Is the vegetation cover comparable in both studies?

*Response: Yes, here, we refer to their samples taken at site NF, but we state that in the sentence.*

P 24 L 277: Debatable that the authors state to see highly dynamic isotope signals.

*Response: We refer to Fig. 3 which shows that.*

P 25 L 290: What exactly is the angle in your case?

*Response: We added in brackets (4°).*

P 25 L 293: gramma: . . .are mainly due to. . .

*Response: Was changed as suggested.*

P 26 L 318: In my opinion, the present study does not really unravel interactions occurring in the soil-plant-atmosphere continuum but adds to process understanding of water fluxes through the soil compartment.

*Response: Here we simply state that the presented soil water isotope data can be used to improve hydrological models.*

**Publication bibliography**

Beyer, M.; Koeniger, P.; Gaj, M.; Hamutoko, J. T.; Wanke, H.; Himmelsbach, T. (2016): A deuterium-based labeling technique for the investigation of rooting depths, water uptake dynamics and unsaturated zone water transport in semiarid environments. In *J Hydrol* 533, pp. 627–643. DOI: 10.1016/j.jhydrol.2015.12.037.

Braun, Hannah (2015): Influence of vegetation on precipitation partitioning and isotopic composition in northern upland catchments. Master's Thesis. Albert-Ludwigs-Universität Freiburg i. Br. Professur für Hydrologie.

Evaristo, Jaivime; Jasechko, Scott; McDonnell, Jeffrey J. (2015): Global separation of plant transpiration from groundwater and streamflow. In *Nature* 525 (7567), pp. 91–94. DOI: 10.1038/nature14983.

Gaj, Marcel; Kaufhold, Stephan; McDonnell, Jeffrey J. (2017): Potential limitation of cryogenic vacuum extractions and spiked experiments. In *Rapid communications in mass spectrometry : RCM. DOI:* 10.1002/rcm.7850.

Geris, Josie; Tetzlaff, Doerthe; Soulsby, Chris (2015): Resistance and resilience to droughts: hydropedological controls on catchment storage and run-off response. In *Hydrol. Process.* 29 (21), pp. 4579–4593. DOI: 10.1002/hyp.10480.

Oerter, Erik; Bowen, Gabriel (2017): In situ monitoring of H and O stable isotopes in soil water reveals ecohydrologic dynamics in managed soil systems. In *Ecohydrol. DOI:* 10.1002/eco.1841.

Sprenger, Matthias; Herbstritt, Barbara; Weiler, Markus (2015): Established methods and new opportunities for pore water stable isotope analysis. In *Hydrol. Process.* 29 (25), pp. 5174–5192. DOI: 10.1002/hyp.10643.

Sprenger, Matthias; Leistert, Hannes; Gimbel, Katharina; Weiler, Markus (2016): Illuminating hydrological processes at the soil-vegetation-atmosphere interface with water stable isotopes. In *Rev. Geophys.* 54 (3), pp. 674–704. DOI: 10.1002/2015RG000515.

Sprenger, Matthias; Tetzlaff, Doerthe; Soulsby, Chris (2017a): No influence of CO2 on stable isotope analyses of soil waters with OA-ICOS. In *Rapid communications in mass spectrometry : RCM* 31 (5), pp. 430–436. DOI: 10.1002/rcm.7815.

Sprenger, Matthias; Tetzlaff, Doerthe; Tunaley, C.; Dick, Jonathan; Soulsby, Chris (2017b): Evaporation fractionation in a peatland drainage network affects stream water isotope composition. In *Water Resour Res* 53 (1), pp. 851–866. DOI: 10.1002/2016WR019258.

Volkmann, Till H. M.; Haberer, K.; Gessler, A.; Weiler, M. (2016): High-resolution isotope measurements resolve rapid ecohydrological dynamics at the soil-plant interface. In *New Phytol* 210 (3), pp. 839–849. DOI: 10.1111/nph.13868.

[revised manuscript text omitted]

The soil samples were analysed for their stable isotopic composition ($\delta^2$H and $\delta^{18}$O) according to the direct equilibration method suggested by Wassenaar et al. (2008). The analyses were conducted within one week after the sampling to prevent microbial activity within the bags. The analyses were done by adding dry air to the bags that contained the soil samples, heat sealing the bags and letting the soil water equilibrate with the dry atmosphere in the bag for two days at constant temperature in the laboratory. The same was done in parallel with bags each filled with 10 ml of one of three different standard waters covering the range of the soil water isotopic signals: seawater ($\delta^{18}$O = –0.85 ‰ and $\delta^2$H = –5.1 ‰), Aberdeen tap water ($\delta^{18}$O = –8.59 ‰ and $\delta^2$H = –57.7), condensate of distilled tap water ($\delta^{18}$O = –11.28 ‰ and $\delta^2$H = –71.8) and for the sampling in January Krycklan snow melt ($\delta^{18}$O = –15.36 ‰ and $\delta^2$H = –114.4). After the equilibration over 2 days, the vapour in the headspace of each bag was sampled directly with a needle connected to an off-axis Integrated Cavity Output Spectroscopy (OA-ICOS) (TWIA-45-EP, Los Gatos Research, Inc., San Jose, CA, USA). The $\delta^{18}$O and $\delta^2$H composition was continuously measured over 6 minutes, of which the last 2 minutes, where the water vapour pressure in the cavity was constant (standard deviation < 100 ppm), were used to calculate average values. The standard deviation for the $\delta^{18}$O and $\delta^2$H measurements were usually < 0.25 ‰ and < 0.55 ‰, respectively. The standards, which were treated the same way as the soil samples and measured at the beginning, the middle and the end of each sampling day, were then used for calibration to derive the isotopic composition of liquid soil waters from vapor measurements. For a detailed description of the soil water isotope analyses in the lab of the Northern Rivers Institute at the University of Aberdeen with the direct equilibration method using off-axis Integrated Cavity Output Spectroscopy (OA-ICOS) (triple-water-vapour isotope analyser TWIA-45-EP, Model#: 912-0032-0000, Serial#: 14-0038, Manufactured: 03/2014, Los Gatos Research, Inc., San Jose, CA, USA), we refer to Sprenger et al. (2017a). To assess the precision of the analysis, we derived the standard deviation of in total 81 measurements of the standard a standard water (Aberdeen tap water) sampled along with the soil samples at the beginning, the middle and the end of each of the on 27 days of laboratory analyses over one year. The standard deviation of the standard water analysis was 0.31 ‰ for $\delta^{18}$O values and 1.13 ‰ for $\delta^2$H values. Recently reported potential effects of $CO_2$ on the isotope analysis of vapour with wavelength-scanned cavity ring-down spectroscopy (Gralher et al., 2016) have been shown to not apply to the OA-ICSO that we used, as shown by Sprenger et al. (2017a). Potentially fractionating effects of interactions between soil water and surfaces of clay minerals (Oerter et al., 2014; Gaj et al., 2017a; Gaj et al., 2017b; Newberry et al., 2017) are of minor relevance for our study, since clay contents were low in the sampled soils (Table 1). We can further ensure that the sampled soil volumes always contained much more than 3 g of water as suggested by Hendry et al. (2015).

In addition to the soil water analysis, precipitation at the field site was sampled with an auto sampler on daily basis at the catchment outlet (location shown in Figure 1a). The auto sampler was emptied at least every two weeks and evaporation from the sampling bottles was prevented by adding paraffin. The precipitation isotopic composition ($\delta^2$H and $\delta^{18}$O) was determined with the above mentioned OA-ICOS running in liquid mode with a precision of 0.4 ‰ for

$\delta^2$H values and 0.1 ‰ for $\delta^{18}$O values, as given by the manufacturer.

**Figure 1 Location of the four sampling sites within the Bruntland Burn catchment on (a) a soil map and (b) an aerial photo. The precipitation sampling location is indicated by a blue triangle in (a). (c) The four photos on the right show exemplary soil profiles for the four study sites.**

**2.5  Data analysis**

We calculated the evaporation line (EL) as a regression line through the soil water isotope data of each sampling date in the dual isotope space . The EL is characterized by its slope and intercept with the $\delta^2$H axis. All regressions for EL presented here were significant at the 95 % confidence interval. For each soil water and precipitation sample, we further calculated the line conditioned excess (lc-excess) as a function of the slope ($a =$

7.6) and the intercept ($b = +4.7$ ‰) of the LMWL (Equation 1) as suggested by Landwehr and Coplen (2006):

$$lc - excess = \delta^2H - a \times \delta^{18}O - b \tag{2}$$

The lc-excess describes the deviation of the sample's $\delta^2$H value the LMWL in the dual isotope space (Landwehr et al., 2014), which indicates non-equilibrium kinetic fractionation processes due to evaporation after precipitation. Therefore, the lc-excess is similar to the well-established deuterium-excess (Dansgaard, 1964) that relates the deuterium composition to the global meteoric water line (GMWL). However, we found that lc-excess was advantageous over the deuterium-excess (or single isotope approaches with $\delta^2$H or $\delta^{18}$O) for inferring evaporation fractionation, because the lc-excess of the precipitation input is about 0 ‰ and with relatively little seasonal dynamics, while $\delta^2H$, $\delta^{18}O$, and d-excess can have an intense seasonal variability (Sprenger et al., 2017b). The accuracy for the liquid and soil water isotope analysis result in a precision limit for lc-excess of about 1.1 ‰ and 3.4 ‰, respectively.

To infer dynamics of potential evaporation rates, we estimated potential evapotranspiration (PET) with the Penman-Monteith-Equation adjusted for the Scottish Highlands by Dunn and Mackay (1995). Note that we focus in our study on the PET dynamics and that the absolute values could vary depending on the aerodynamic and roughness parameter of different vegetation covers. We further did not partition PET into evaporation and transpiration fluxes, since PET was primarily used as a proxy for potential soil evaporation rates, and evaporation and transpiration usually show a linear relationship in temperate regions (Renner et al., 2016; Schwärzel et al., 2009). To understand the potential atmospheric drivers for the soil water isotopic composition, we investigated the effect of antecedent conditions. We calculated average values of PET over 30 days prior to each sampling campaign were calculated ($PET_{30}$) to account for the potential soil evaporation dynamics. Additionally, the precipitation sums and the amount weighted isotopic signal of the daily precipitation isotope samples were computed for the over 7 days and 30 days period prior to the sampling was calculated ($P_7$ and $P_{30}$, respectively) to assess the mixing processes between precipitation input and soil water. Weekly and monthly averages were chosen to see if the relatively young water input or the average over the last month relate differently to the observed soil water isotopic signal.

We estimated the evaporative water losses $f$ and evaporation/input ratios ($E/I$) based on the Craig-Gordon model (Craig and Gordon, 1965) with the *Hydrocalculator* provided by Skrzypek et al. (2015). For a detailed description of the calculations, we refer to Skrzypek et al. (2015) and will only briefly introduce the main characteristics of the approach and assumptions we used in our application. The estimate of the evaporative losses $f$ [%] is based on the Craig-Gordon model (Craig and Gordon, 1965) and formulations introduced by Gonfiantini (1986) and adapted in the *Hydrocalculator* for isotope mass balance as follows:

$$f = 1 - \left[\frac{(\delta_S - \delta^*)}{(\delta_P - \delta^*)}\right]^m \tag{3}$$

For our estimates of $f$, wWith defined $\delta_S$ defined as the average weighted isotopic signal of the soil water in the upper 10 cm. The upper 10 cm were chosen, because this was the depth with the highest evaporation signal in the soil water isotopes (see results section) and where most evaporation are usually observed in laboratory experiments (Or et al., 2013)., as shown in the results section. The sampled soil water isotope signal $\delta_{SW}$ was corrected for the seasonality of the input signal by the average isotopic signal of $P_{30}$ as $\delta_S = \delta_P + (\delta_{SW} - \delta_{P30})$, $\delta_P$ was defined as the isotopic signal of the original water source by calculating the intercept between the evaporation line of the soil water isotope data in the dual-isotope space (see Figure S 2) and the LMWL according to Javaux et al. (2016). in the long-term precipitation input (-61 ‰ for $\delta^2H$ and -8.5 ‰ for $\delta^{18}O$), $\delta^*$ as-is the limiting isotopic enrichment factor, and $m$ as-is the enrichment slope, both described by Gibson and Reid (2014) (Equation (8) and (9) therein). $\delta^*$ is a function of the air humidity $h$, the isotopic composition of the ambient air $\delta_A$, and a total enrichment factor $\varepsilon$ (Gat and Levy, 1978). For humidity, we averaged over 30 days prior to each soil water sampling date the measured humidity at a meteorological station less than 800 m away from the study sites $h_{30}$. $\delta_A$ was derived as function from the weighted average precipitation input of the 30 days prior to the soil sampling $\delta_{P30}$ and the equilibrium isotope fractionation factor $\varepsilon^+$. (Gibson et al., 2008),  with the latter  depending on the temperature as given by Horita and Wesolowski (1994). We used air temperature data from the aforementioned meteorological station and computed values averaged over 30 days prior to soil water sampling $T_{30}$. The total enrichment factor $\varepsilon$ is the sum of the equilibrium isotope fractionation factor $\varepsilon^+$ and the kinetic isotope fractionation factor $\varepsilon_\kappa$, which is a function of $h_{30}$, the exponent of the diffusion coefficient ratio $n$ and a kinetic fractionation constant, which has a value of 28.4 ‰ for $\delta^{18}O$ and 25.0 for $\delta^2H$ (Gonfiantini, 1986). Here, we define $n = 1$ in accordance to Barnes and Allison (1983), representing diffusional transport in soil pores. 
[revised manuscript text omitted]

**4. Discussion**

**4.1 Mixing of precipitation input and the critical zone water storage**

Our uniquely detailed data set from 11 sampling campaigns over one year revealed the isotopic response of soil water
to variable precipitation inputs. An understanding of the isotopic variability of the precipitation input signal is crucial
for the interpretation of the soil water isotopes in terms of soil evaporation dynamics at the soil-plant-atmosphere
interface. Therefore, we will first discuss how mixing of the infiltrating precipitation within the topsoil provides the
basis for the isotopic enrichment in the soil due to evaporative losses.

Given that the soil at the four sites consists of similar texture, it is not too surprising that the SW generally responded
similarly to the seasonal P $\delta^2$H and $\delta^{18}$O input signal. However, we would expect differences in the soil physical
properties due to the differences in the organic matter contents. Our data showed that higher organic matter resulted
in higher GWCs at the sites. That means that the soil water storage is higher for the sites with higher organic material,
which is in line with several other studies (e.g., Hudson, 1994).

While this relationship between GWC and LOI persisted independently of how many dry days occurred prior to the
soil sampling, LOI showed a relationship with lc-excess depending on the dryness. This suggests that during dry
periods, the external atmospheric drivers such as evaporative demand may have a high influence on the SW lc-excess.
In contrast, during wet periods, the soil water storage capacities – here, mainly controlled by organic material -– gain
importance. With increasing new usually unfractionated precipitation input (lc-excess close to zero), the sites with
higher organic matter content have a different SW lc-excess than the sites with lower organic matter content, because
there is a higher mixing volume for the sites with the high LOI compared to the soils with lower LOI (Figure 6Figure
7). Hence, future and current changes in the organic matter content of soils with high organic matter due to, for
example, land use or climate changes (Foley et al., 2005; Rees et al., 2011), would have an impact on the mixing of
event and pre-event water in the podzols.

Interestingly, the highest variability of the $\delta^2$H values in the soil water was found for the sampling after the intense
rainfalls in January 2016 (Figure 4Figure 5a, Figure 7Figure 9a), which indicates that the unsaturated zone was not
homogeneously wetted by the exceptionally high precipitation input. The GWC corroborates this, because its
variability for the sampling campaign in January was not lower than for other sampling days (Figure 9b). This suggests
that bypass flow occurred and the stored soil water was not necessarily well-mixed. However, the significantly lower

$\delta^2$H values in the top 5 cm indicate that, despite the potential occurrence of preferential flow, most of the depleted precipitation input was stored in the very top soil and the more enriched soil water from autumn was partly mixed with and partly replaced by the event water. The sampling in January was  also special, because the GWC was about as high in the soils beneath Scots pine than in the soils beneath heather. Hence, differences in GWC between soils beneath the two vegetation types during the vegetation period as discussed below would stem from different transpiration and soil evaporation rates and not from differences in the soil water storage capacities.

The higher variability of the SW $\delta^2$H values beneath Scots pine compared to the SW beneath heather (Figure 9

) cannot be explained by differences of the throughfall isotopic signal, since they are minor for the two vegetation types (Braun, 2015). The higher variability in the isotopic signal therefore indicates that flow paths are generally more variable in the forest soils. This is probably due to preferential flows via larger macropores formed by tree roots that also reach deeper than the shallower roots of heather. While the high variability of SW $\delta^2$H values indicate preferential flow paths, there was also a clear signal that much of the input water percolates through the pore matrix, since we see that the depleted winter precipitation signal is still evident at -20 cm soil depth during the March sampling campaign (Figure 7a).

The relatively high soil water storage volumes in the podzols result in a damping of the $\delta^2$H signal in the SW compared to the precipitation input. There was also a more damped isotopic signal with soil depth due to increased mixing and cumulatively larger soil water storage. Because of this damping effect, the differences between the sites (Figure

5a, Figure 9a) and sampling times (Figure 7) decreased with depth.

Our data support the findings of Geris et al. (2015b), who showed - with soil water samples extracted with suction lysimeters - a more damped signal in deeper layers of the soils beneath heather than beneath Scots pine. However, we did not see a more delayed response beneath Scots pine compared to podzols beneath heather as observed by Geris et al. (2015b). This discrepancy could arise from the fortnightly sampling frequency by Geris et al. (2015b) and monthly sampling frequency in the current study, as well the effect of an unusually dry summer in former study.

In contrast to Geris et al. (2015b), who sampled the soil-vegetation units with duplicates, our sampling design with five replicates allowed for a clearer assessment of the spatial heterogeneity of the subsurface.

For example, the standard deviation of the SW $\delta^2$H and $\delta^{18}$O values for the -5 cm depth increments at each site was –

for the entirely sampled upper 20 cm of soil – always higher than the measurement accuracy of 1.13 ‰ and 0.31 ‰, respectively. At Bruntland Burn – as in most Northern temperate and boreal biomes (Jackson et al., 1996) – topsoils contain almost all the root biomass. For the interpretation of potential sources of root water uptake this means that the uncertainty of the potential water source signal – caused by the heterogeneous isotopic composition at particular depths within the rooting zone - is higher than the measurement errors . Our field measurements underline, therefore, the need for an improved spatial resolution of soil water sampling when studying root water uptake patterns with stable isotopes, as recently called for by Berry 
[revised manuscript text omitted]

333 The evaporation estimates are limited to the period of highest evaporation fluxes in the Bruntland Burn and are likely
334 to be lower than what we present, since we relate the isotopic enrichment to the long term average input signal in our
335 calculations. However, the assumption of relating the enrichment to the average input seems to be valid, since we have
336 seen the steady state balance of unfractionated input and kinetic fractionation due soil evaporation during the summer
337 months. Nevertheless, detailed estimates of the evaporative fluxes require transient modelling of all water fluxes (i.e.,
338 precipitation, transpiration, evaporation, recharge) and their isotopic composition, which will be subject of future
339 work. Potential transient numerical modelling approaches that account for isotopic fractionation of the soil water
340 isotopes are available (Braud et al., 2005; Rothfuss et al., 2012; Mueller et al., 2014).

341 The frequently measured soil water isotope data we have presented and the inherent evaporation signal can help to
342 calibrate or benchmark the representation of soil-vegetation-atmosphere interactions in tracer aided hydrological
343 modelling from the plot (Rothfuss et al., 2012; Sprenger et al., 2016b) to the catchment scale (Soulsby et al., 2015;
344 van Huijgevoort et al., 2016). So far, the isotopic composition of mobile water has been used to better constrain semi-
345 distributed models (Birkel et al., 2014; van Huijgevoort et al., 2016). Including the isotopes of the bulk soil water,
346 could allow for an improved conceptualization of the soil-plant-atmosphere interface, which is crucial for an adequate
347 representation of evaporation and transpiration. Especially the marked differences of the evaporation signal within the
348 first few centimetres of the soil depth will significantly affect the age distribution of transpiration (Sprenger et al.,
349 2016c), evaporation (Soulsby et al., 2016a) or evapotranspiration (Harman, 2015; Queloz et al., 2015; van Huijgevoort
350 et al., 2016).

**5. Conclusions**

Our study provides a unique insight into the soil water stable isotope dynamics in podzolic soils under different vegetation types at a northern latitude site. We showed that, despite a relatively low energy environment in the Scottish

Highlands, the temporal variability of soil water isotopic enrichment was driven by changes of soil evaporation over the year. The monthly frequency of the soil water isotope sampling corroborates the importance of covering transition periods in a climate with seasonally variable isotopic precipitation input signals and evaporative output fluxes. Missing sampling these periods of higher temporal variability in spring and autumn could pose problems when referring plant water isotopes to potential soil water sources or when using soil water isotopic information for calibrating hydrological models. Especially the delayed response of the soil water lc-excess to evaporation (hysteresis effect) provides valuable insight into how unfractionated precipitation input and kinetic fractionation due to soil evaporation both affect mixing processes in the upper layer of the critical zone. The fact that the evaporation signal generally disappears within the first 15 cm of the soil profile emphasises the importance and the spatial scale of the processes taking place at the soil- vegetation-atmosphere interface of the critical zone.

The vegetation type played a significant role for the evaporation losses, with a generally higher soil evaporation signal in the soil water isotopes beneath Scots pine than beneath heather. Notably, these differences - as indicated in the soil water lc-excess - remain limited to the top 15 cm of soil. The vegetation cover directly affected the evaporation losses with soil evaporation being twice as high as beneath Scots pine (10 % of infiltrating water) compared to soils beneath heather (5 % of the infiltrating water) during the growing season.

The presented soil water isotope data covering the seasonal dynamics of exceptionally high sampling frequency in time (monthly) andat high space spatial resolution (5 cm increments at four locations) will allow us to test efficiencies of soil physical models in simulating the water flow and transport in the critical zone (as called by for Vereecken (2016)). , but The data can further provide also a basis to benchmark hydrological models to realizefor a more realistic representation of the soil-vegetation interactionscelerities and velocities when simulating water fluxes and their ages within catchments (McDonnell and Beven, 2014).

**Appendices**

[Figure]

**Figure S 1 Spatial variability of (a) δ²H and (b) δ¹⁸O in soil water and the effects of taking several samples in parallel to get**
**average values (blue dots and primary y-axis) for the samples if only having one sample (1 on the x-axis) or averaging over**
**2 to 8 samples (2 to 8 on the x-axis). Red dots indicate the standard deviation of the different combinations and the red**
**dotted line shows the precision of the isotope analysis. All 8 samples were taken in the upper 5 cm of soil within 10 m**
**distance to each other.**

[Figure]

**Figure S 2 Dual isotope plots for each sampling day showing the pore water isotopic composition and the isotopic signal in the precipitation averaged over the 7 days (black stars), 14 days (grey stars), and 30 days (light grey stars) prior to the sampling date. Colors indicate the sampling site and marker indicate the sampling depth. Solid line shows the GMWL and the dotted line represents the local meteoric water line (LWM: $\delta^2H = \delta^{18}O \times 7.6 + 4.7$ ‰).**

**Author contribution**

D.T., C.S. and M.S. established the sampling design together, M.S. conducted the field and laboratory work and statistical analysis, all authors were involved in the data interpretation, M.S. prepared the manuscript with contributions from both co-authors.

**Competing interests**

The authors declare that they have no conflict of interest.

**Data availability**

The data is available upon request.

**Acknowledgements**

We are thankful for the support by Audrey Innes during all laboratory work. We further thank Jonathan Dick for running the isotope analysis of precipitation samples and Annette C. Raffan for her support in the soil texture analysis. We would also like to thank the European Research Council (ERC, project GA 335910 VeWa) for funding.

[revised manuscript text omitted]